# Discrete Survival Knowledge Distillation for Competing Risks Analysis

Feiyang Deng [1]   Lingfeng Luo [1]   Di Wang [2]   Qinmengge Li [1]   Lingxuan Kong [1]   Kevin He [1]

## Abstract

Accurate prediction in survival analysis with competing risks is challenged by rare event rates and limited effective sample sizes. Knowledge distillation offers a promising way to transfer information from an external teacher to improve a local student, but existing methods are overwhelmingly developed for uncensored outcomes and do not directly extend to survival analysis, where censored observations provide only partial information. Moreover, prior work often assumes that teacher and student share identical outcome definitions, whereas in competing risks settings, they may differ in outcome granularity and event definitions, further complicating knowledge transfer. To address these gaps, we propose DiSKD (Discrete Survival Knowledge Distillation), a deep learning framework for discrete-time competing risks that integrates teacher predictions via a cause-specific, time-dependent Kullback–Leibler divergence. DiSKD enables flexible and privacy-conscious transfer without requiring raw data sharing, remains robust to model misspecification or outcome-definition heterogeneity, and adaptively weights teacher guidance by emphasizing compatible teachers while down-weighting less relevant ones. Simulation studies and real-world applications demonstrate improved discrimination and calibration.

## 1. Introduction

Deep learning approaches have emerged as promising tools for survival analysis. For example, discrete-time survival models using neural networks have gained traction in clinical applications (Gensheimer & Narasimhan, 2019; Kvamme & Borgan, 2021; Wiegrebe et al., 2024). More re-cent work incorporates attention mechanisms for improved temporal modeling (Hu et al., 2021; Zisser & Aran, 2024), while methods like DeepHit (Lee et al., 2018a) extend this framework to competing risks by directly learning probability mass functions over event types and time points. Despite rapid methodological advances, the success of deep learning typically relies on large-scale datasets. In survival analysis, however, deep learning faces hurdles such as rare events, heavy censoring, small sample sizes, high dimensionality, and low signal-to-noise ratios. These issues are amplified in competing risks settings, where censoring and sparse cause-specific failures compound these challenges and further limit information (Liu et al., 2014).

Our endeavor is motivated by kidney transplantation, widely regarded as the preferred treatment for patients with renal failure (Wolfe et al., 1999). However, demand for kidney transplantation far exceeds available supply. Currently, approximately 93,000 patients remain on the U.S. kidney transplant waiting list. Due to organ scarcity, the median waiting time is about four years, during which roughly 5% of patients die each year without receiving a transplant. Paradoxically, 30% of donated kidneys (over 8,000 annually) are discarded, often due to concerns about donor quality (Marrero et al., 2017). In contrast, kidney discard rates in Europe are typically below 10%. The dilemma is that accepting lower-quality donors can result in better survival than continued waiting, but may also result in worse post-transplant outcomes. Patients therefore face a trade-off: they can select a higher-quality organ threshold and stay on the waiting list longer, which may increase the risk of dying, or select a lower-quality organ threshold and receive a transplant sooner but possibly experience earlier graft failure. Yet there are no clinical practice guidelines or consensus recommendations that provide patient-specific decision support about donor quality and associated outcomes.

Despite urgent needs, accurate risk prediction for organ transplantation requires addressing several challenges. First, post-transplant prediction is inherently a competing-risks problem, involving graft failure and patient death with a functioning graft as two mutually exclusive terminal outcomes that provide distinct but interrelated information about transplant prognosis. Risk prediction with such competing events is challenged by heavily censored follow-ups and strong cross-cause dependence. Second, kidney trans-

---

[1]Department of Biostatistics, University of Michigan, Ann Arbor, MI, USA [2]Division of Biostatistics, Data Science Institute, Medical College of Wisconsin, Milwaukee, WI, USA. Correspondence to: Kevin He <kevinhe@umich.edu>.

*Proceedings of the $43^{rd}$ International Conference on Machine Learning*, Seoul, South Korea. PMLR 306, 2026. Copyright 2026 by the author(s).

plant data exhibit substantial temporal and geographic heterogeneity due to overlapping disruptions from the COVID-19 pandemic and major policy reforms (see Figure 3a in Section 5.5 and Figure 8 in Appendix E.3.1). In particular, the nationwide implementation of the KAS250 policy in March 2021 (which removed donation service areas and expanded organ sharing to a 250-mile radius) coincided with the recovery from the first wave of the pandemic (Cron et al., 2023). During this period, transplant center activity fluctuated, recipient mortality rose, and donor profiles shifted toward higher-risk categories, creating distributional shifts in post-transplant risk patterns. Consequently, post-KAS250 prediction is essential for policy assessment and clinical care, yet post-KAS250 cohorts are small, short-term, and event-sparse, whereas larger pre- and mid-pandemic data may no longer represent current practice. Third, patient privacy constraints often restrict sharing individual-level historical data; instead, only summary information from published models can be exchanged. Finally, integration is complicated by heterogeneity across models and data sources in covariate definitions, feature sets, and endpoint formulations. For instance, the widely used post-transplant model (Snyder et al., 2016) was developed using overall failure time without distinguishing competing risks, limiting compatibility with more granular cause-specific models.

Our proposed solution was motivated by knowledge distillation, which trains internal "student" models by transferring information from external "teacher" models (Hinton et al., 2015; Gou et al., 2021). This distillation allows the student model to leverage external knowledge and improve generalization and robustness under data constraints. Although successful for classification tasks (Ma et al., 2021; Ranjbari & Arslanturk, 2023), these methods are not directly applicable to censored time-to-event data. In particular, classical Kullback–Leibler (KL) divergence-based distillation objectives assume that each training subject has a well-defined target distribution so the student can be trained to mimic the teacher's predicted distribution. For uncensored observations in survival analysis, the event time is observed, yielding a likelihood contribution at that time and making teacher–student alignment conceptually feasible. For censored observations, however, the data only reveal that the failure time exceeds the censoring time, and thus there is no available event-time probability mass (or density) function to match. This mismatch makes standard distribution-based distillation ill-defined for censored individuals.

On the other hand, although transfer learning methods (Chen et al., 2021; Sheng et al., 2021; Cheng et al., 2023; Wang et al., 2025) have been proposed for survival data integration, these techniques are largely rooted in classical regression frameworks. They often assume linear covariate effects and may not extend naturally to nonlinear or high-capacity deep learning models. In addition, approaches such as Li et al.

(2022) and Tian & Feng (2023), which penalize coefficient differences between external and internal models, implicitly assume a shared parameterization and covariate space. This becomes problematic in deep neural networks, where parameters rarely correspond one-to-one and feature representations can differ due to preprocessing, embeddings, or learned transformations. These incompatibilities make direct coefficient alignment infeasible.

To address these challenges, we propose a discrete-time knowledge distillation method (DiSKD) that applies to both standard time-to-event and competing-risks settings. The framework offers the following advantages: (1) it leverages deep neural networks to model complex nonlinear relationships, covariate interactions, and temporal dynamics, improving predictive accuracy; (2) it adaptively weights information from teachers, emphasizing compatible sources and down-weighting less relevant ones; (3) it is robust to model misspecification and accommodates diverse teachers, including Cox models, discrete-time survival models, and deep learning approaches, as long as they provide predicted probabilities; (4) it allows partial overlap in covariate sets between teacher and student models; (5) it accommodates outcome-definition heterogeneity between teacher and student models, such as when the student model targets competing risks while teachers report only aggregated (overall) failures; (6) it relies solely on teacher predictions rather than raw data or model parameters, respecting privacy constraints; and (7) it incurs computational cost comparable to standard deep learning procedures, making it scalable to large-scale settings.

**Code Availability.** A companion repository with the DiSKD implementation and scripts for selected experiments is available at: `https://github.com/yatoka233/DiscreteSurvKD`.

## 2. Method

### 2.1. Internal Discrete Competing Risks Model

Let $D_i$ and $C_i$ be the failure and censoring times, respectively, for subject $i$ in the internal cohort, $i = 1, \ldots, n$, where $n$ is the sample size. Let $\Delta_i \in \{1, \ldots, J\}$ be the indicator of the cause of failure, where $J$ is the total number of competing risks. Let $\tau_1, \ldots, \tau_K$ be the discrete follow-up times. Let $\boldsymbol{Z}_i$ be a covariate vector for subject $i$. Assume that $D_i$ is independent of $C_i$, conditional on $\boldsymbol{Z}_i$. The cause-specific discrete hazard (Kalbfleisch & Prentice, 2002) for cause $j$ at time $\tau_k$, $k = 1, \ldots, K$, given covariates $\boldsymbol{Z}_i$, is defined as:

$$\lambda_j(\tau_k; \boldsymbol{Z}_i) = P(D_i = \tau_k, \Delta_i = j \mid D_i \geq \tau_k, \boldsymbol{Z}_i),$$

which represents the probability that subject $i$ experiences an event from cause $j$ exactly at time $\tau_k$, conditional on survival

to $\tau_k$. Specifically, we consider the following formulation:

$$\lambda_j(\tau_k; \mathbf{Z}_i) = \frac{\exp\{r_j(\tau_k; \mathbf{Z}_i)\}}{1 + \sum_{j^*=1}^{J} \exp\{r_{j^*}(\tau_k; \mathbf{Z}_i)\}}, \, j = 1, \ldots, J,$$

where $r_j(\tau_k; \mathbf{Z}_i)$, $j = 1, \ldots, J$, are real-valued functions of time and covariates. Define $\mathbf{r} = (r_1, \ldots, r_J)^{\top}$. Let $\delta_{ik}^{(j)} = I(\min(D_i, C_i) = \tau_k, \Delta_i = j)$ be the event indicator for cause $j$ at time $\tau_k$, and $Y_{ik} = I(\min(D_i, C_i) \geq \tau_k)$ be the at-risk indicator for subject $i$ at $\tau_k$. The corresponding log-likelihood of the internal cohort is:

$$\ell(\mathbf{r}) = \sum_{i=1}^{n} \sum_{k=1}^{K} Y_{ik} \left[ \sum_{j=1}^{J} \delta_{ik}^{(j)} r_j(\tau_k; \mathbf{Z}_i) \right.$$
$$\left. - \log\left(1 + \sum_{j=1}^{J} \exp\{r_j(\tau_k; \mathbf{Z}_i)\}\right) \right]. \quad (1)$$

### 2.2. Teacher Predictions

Suppose there is a teacher model (e.g. a discrete failure-time model or a Cox model) that takes $\mathbf{Z}_i$, or a subset of its covariates, as input and outputs predicted cause-specific hazards:

$$\tilde{\lambda}_j(\tau_k; \mathbf{Z}_i) = \tilde{P}(D_i = \tau_k, \Delta_i = j | D_i \geq \tau_k, \mathbf{Z}_i),$$

where $\tilde{P}$ is the distribution of the teacher model.

### 2.3. Distilled Competing Risks Model

To quantify heterogeneity between student and teacher, we leverage two observations: First, the likelihood of the discrete cause-specific hazards model, as in (1), is fully determined by the cause-specific hazards, which are conditional probabilities; Second, in a discrete-time competing risks setup, each subject contributes a sequence of multinomial outcomes over time: at each interval $\tau_k$, one of the $J$ causes may occur (or none, if the subject survives), where the event probabilities are given by the cause-specific hazards and the no-event probability. These insights motivate us to define a cause-specific time-dependent KL divergence between the distribution implied by the teacher $\tilde{P}$ and that of the student $P_{\mathbf{r}}$,

$$d(\tilde{P}\|P_{\mathbf{r}}; \tau_k, \mathbf{Z}_i) = \sum_{j=1}^{J} \tilde{\lambda}_j(\tau_k; \mathbf{Z}_i) \log\left(\frac{\tilde{\lambda}_j(\tau_k; \mathbf{Z}_i)}{\lambda_j(\tau_k; \mathbf{Z}_i)}\right)$$
$$+ \{1 - \tilde{\lambda}(\tau_k; \mathbf{Z}_i)\} \log\left(\frac{1 - \tilde{\lambda}(\tau_k; \mathbf{Z}_i)}{1 - \lambda(\tau_k; \mathbf{Z}_i)}\right), \quad (2)$$

where $\lambda(\tau_k; \mathbf{Z}_i) = \sum_{j=1}^{J} \lambda_j(\tau_k; \mathbf{Z}_i)$ and $\tilde{\lambda}(\tau_k; \mathbf{Z}_i) = \sum_{j=1}^{J} \tilde{\lambda}_j(\tau_k; \mathbf{Z}_i)$ are the overall hazards across all causes. To accumulate information from longitudinal multinomial

outcomes across subjects in the internal data, the accumulated KL divergence is given by,

$$\mathrm{D}(\tilde{P}\|P_{\mathbf{r}}) = \sum_{i=1}^{n} \sum_{k=1}^{K} Y_{ik} d(\tilde{P}\|P_{\mathbf{r}}; \tau_k, \mathbf{Z}_i), \quad (3)$$

which measures the discrepancy between the student distribution and the teacher guidance over all at-risk intervals. To balance the trade-off between the teacher and the internal data, we aim to maximize the log-likelihood of the internal data, while simultaneously keeping the KL divergence small. This is achieved by maximizing a penalized log-likelihood

$$\ell_{\eta}(\mathbf{r}) = \ell(\mathbf{r}) - \eta \, \mathrm{D}(\tilde{P}\|P_{\mathbf{r}}), \quad (4)$$

where $\eta \geq 0$ is a hyperparameter that tunes trust in teacher predictions, giving larger weight to more comparable ones and down-weighting less relevant ones. In the extreme case $\eta = 0$, $\ell_{\eta}(\mathbf{r})$ reduces to the log-likelihood, defined in (1), based only on internal data. When $\eta \to \infty$, maximizing $\ell_{\eta}(\mathbf{r})$ is equivalent to minimizing $\mathrm{D}(\tilde{P}\|P_{\mathbf{r}})$, forcing the model to replicate teacher predictions.

**Temperature–Adjusted KL for Competing Risks.** The same formulation also allows a temperature-adjusted KL component. In classical knowledge distillation, temperature scaling softens the teacher's predictive distribution and reveals relative similarities among non-target classes (Hinton et al., 2015). In competing risks, this is useful because non-realized causes may still carry information about cross-cause dependence, such as shared latent health status or common etiologic pathways, that may be weakly identified from limited internal data alone. To incorporate this information without changing the internal likelihood, we apply temperature scaling only to the KL component. For $T > 0$, define the temperature-scaled teacher distribution

$$\tilde{\lambda}_j^{(T)}(\tau_k; \mathbf{Z}_i) = \frac{\{\tilde{\lambda}_j(\tau_k; \mathbf{Z}_i)\}^{\frac{1}{T}}}{\{1 - \tilde{\lambda}(\tau_k; \mathbf{Z}_i)\}^{\frac{1}{T}} + \sum_{l=1}^{J} \{\tilde{\lambda}_l(\tau_k; \mathbf{Z}_i)\}^{\frac{1}{T}}},$$

and define $\lambda_j^{(T)}(\tau_k; \mathbf{Z}_i)$ similarly for the student. The temperature-adjusted cause-specific KL divergence is

$$d_T(\tilde{P}\|P_{\mathbf{r}}; \tau_k, \mathbf{Z}_i) = \sum_{j=1}^{J} \tilde{\lambda}_j^{(T)}(\tau_k; \mathbf{Z}_i) \log\left(\frac{\tilde{\lambda}_j^{(T)}(\tau_k; \mathbf{Z}_i)}{\lambda_j^{(T)}(\tau_k; \mathbf{Z}_i)}\right)$$
$$+ \{1 - \tilde{\lambda}^{(T)}(\tau_k; \mathbf{Z}_i)\} \log\left(\frac{1 - \tilde{\lambda}^{(T)}(\tau_k; \mathbf{Z}_i)}{1 - \lambda^{(T)}(\tau_k; \mathbf{Z}_i)}\right),$$

where $\tilde{\lambda}^{(T)}(\tau_k; \mathbf{Z}_i) = \sum_{j=1}^{J} \tilde{\lambda}_j^{(T)}(\tau_k; \mathbf{Z}_i)$ and similarly for $\lambda^{(T)}(\tau_k; \mathbf{Z}_i)$. The accumulated divergence $\mathrm{D}_T(\tilde{P}\|P_{\mathbf{r}})$ is defined by replacing $d$ with $d_T$ in (3), and the corresponding objective is obtained by replacing $\mathrm{D}$ with $\mathrm{D}_T$ in (4). When $T = 1$, this reduces to the original KL divergence in (2).

For $T > 1$, the softened teacher distribution places greater emphasis on the relative contrast among non-target risks, providing a mechanism for borrowing information about similarity structures across competing risks. In practice, $T$ is tuned jointly with $\eta$ via cross-validation. Details on this selection procedure are provided in Section 5.2.

### 2.4. Estimation with Neural Networks

The propositions below provide two equivalent views of the proposed objective. For clarity, they are stated for the default KL objective in (4), corresponding to $T = 1$; the temperature-adjusted version is obtained by replacing D with $D_T$ in the KL component while keeping the internal likelihood unchanged.

Proposition 2.1 demonstrates that the penalized log-likelihood, $\ell_\eta(\boldsymbol{r})$, minimizes a convex combination of KL divergences between the working model and two extremes: one based solely on the internal cohort and the other solely on the teacher.

**Proposition 2.1.** *Let $P_n$ be the empirical probability measure on the internal cohort. For a given $\eta \geq 0$, the model $P_{\boldsymbol{r}}$ minimizing $(1-\alpha)\mathrm{D}(P_n\|P_{\boldsymbol{r}}) + \alpha\mathrm{D}(\tilde{P}\|P_{\boldsymbol{r}})$ is the distilled model in (4), where $\alpha = \eta/(1+\eta)$.*

Thus, the minimizer seeks a distribution close to both the empirical internal distribution $P_n$ and the teacher distribution $\tilde{P}$, with the relative contribution controlled by the tuning parameter. At the level of unconstrained interval-level categorical probabilities, the objective is convex in the student distribution, and the corresponding optimal interval-level distribution is unique when it lies in the model class. With a neural-network parameterization, however, the optimization problem is generally nonconvex, and uniqueness in parameter space is not guaranteed. Proposition 2.2 below shows that the proposed method provides a structure that is fully compatible with the original log-likelihood in (1).

**Proposition 2.2.** *Ignoring constant terms, the penalized log-likelihood in (4) is proportional to*

$$\ell_\eta(\boldsymbol{r}) \propto \sum_{i=1}^{n} \sum_{k=1}^{K} Y_{ik} \left[ \sum_{j=1}^{J} \frac{\delta_{ik}^{(j)} + \eta\tilde{\lambda}_j(\tau_k; \boldsymbol{Z}_i)}{1 + \eta} r_j(\tau_k; \boldsymbol{Z}_i) \right. $$
$$\left. - \log\Big(1 + \sum_{j=1}^{J} \exp\{r_j(\tau_k; \boldsymbol{Z}_i)\}\Big) \right]. \quad (5)$$

Proposition 2.2 shows that DiSKD preserves the same multinomial likelihood structure as the internal discrete-time competing risks model, but replaces the observed event indicator with a convex combination of the empirical label and the teacher-predicted hazard. This provides a direct implementation route: the same neural hazard model can be trained with teacher-augmented interval-level targets.

To illustrate flexibility, we instantiate the hazard network with four neural architectures, originally developed for single-risk settings but extended here to competing risks: (1) Nnet-Survival (Gensheimer & Narasimhan, 2019): a lightweight Multilayer Perceptron (MLP) enforcing proportional hazards; (2) Logistic-Hazard (Kvamme & Borgan, 2021): a feed-forward MLP outputting the hazard vector with time-dependent effects; (3) Time-Embedding: concatenates covariate and learnable time embeddings, processed by feed-forward layers for interval-specific outputs; and (4) Transformer (Hu et al., 2021): integrates covariate and time embeddings in a multi-head self-attention structure to capture temporal dependencies and complex interactions.

**Cumulative Incidence Function (CIF).** Given the fitted cause-specific hazards, the CIF (Gray, 1988), for each cause $j$, can be calculated as:

$$F_j(\tau_k; \boldsymbol{Z}_i) = P(D_i \leq \tau_k,\, \Delta_i = j \mid \boldsymbol{Z}_i)$$
$$= \sum_{l=1}^{k} \lambda_j(\tau_l; \boldsymbol{Z}_i) \prod_{l^*=1}^{l-1} \big\{ 1 - \lambda(\tau_{l^*}; \boldsymbol{Z}_i) \big\}, \quad (6)$$

which quantifies absolute risk for a specific event type in the presence of competing events.

## 3. Theoretical Properties

We provide a local excess-risk expansion to clarify the role of the KL penalty. The result is not intended as a universal improvement guarantee; rather, it formalizes the bias–variance trade-off underlying DiSKD. Teacher guidance can reduce variance when the internal cohort is small or event-sparse, but it can also introduce bias when the teacher is poorly aligned with the target population. This motivates selecting the distillation weight using internal validation data.

For notational simplicity, we state the result for a discrete-time survival model with one event type. The competing-risks case follows by replacing the Bernoulli interval outcome with the corresponding multinomial interval-level outcome distribution. Full assumptions and proofs are given in Appendix A.4.

Let $\boldsymbol{\theta}$ denote the parameters of a differentiable neural hazard model with conditional hazard

$$\lambda_{\boldsymbol{\theta}}(\tau_k; \boldsymbol{Z}) = \sigma\{r_{\boldsymbol{\theta}}(\tau_k; \boldsymbol{Z})\}.$$

Suppose the teacher provides hazards $\tilde{\lambda}(\tau_k; \boldsymbol{Z})$ on the same time grid. Define the empirical internal risk, empirical KL criterion, and KL-integrated objective as

$$R_n(\boldsymbol{\theta}) = -\frac{1}{n}\sum_{i=1}^{n} \ell_i(\boldsymbol{\theta}), \quad Q_n(\boldsymbol{\theta}) = \frac{1}{n}\sum_{i=1}^{n} q_i(\boldsymbol{\theta}),$$

where $q_i(\boldsymbol{\theta})$ is the accumulated interval-level KL divergence between the teacher and student hazards. Then the KL-integrated objective is

$$M_{n,\eta}(\boldsymbol{\theta}) = R_n(\boldsymbol{\theta}) + \eta Q_n(\boldsymbol{\theta}).$$

Let $R(\boldsymbol{\theta}) = E\{R_n(\boldsymbol{\theta})\}$ and $Q(\boldsymbol{\theta}) = E\{Q_n(\boldsymbol{\theta})\}$. Let $\boldsymbol{\theta}^\star$ minimize the internal population risk $R(\boldsymbol{\theta})$, and define

$$\boldsymbol{H} = \nabla^2 R(\boldsymbol{\theta}^\star), \quad \boldsymbol{b} = \nabla Q(\boldsymbol{\theta}^\star), \quad \boldsymbol{G} = \nabla^2 Q(\boldsymbol{\theta}^\star),$$

with $\boldsymbol{\Sigma}$ denoting the asymptotic covariance of the internal score. Here $\boldsymbol{b}$ measures local teacher–target discrepancy: it is small when the teacher-induced hazards align with the target population, and large when the KL term pulls the student toward an incompatible direction.

**Theorem 3.1** (Local excess-risk expansion). *Assume the local regularity conditions in Appendix A.4. Let $\widehat{\boldsymbol{\theta}}_{\eta_n}$ be a local empirical minimizer of $M_{n,\eta_n}(\boldsymbol{\theta})$ with $\eta_n \to 0$, and define*

$$\boldsymbol{A}_\eta = (\boldsymbol{H} + \eta\boldsymbol{G})^{-1}.$$

*Then*

$$E\left[ R(\widehat{\boldsymbol{\theta}}_{\eta_n}) - R(\boldsymbol{\theta}^\star) \right] = \frac{1}{2n} \operatorname{tr}\left( \boldsymbol{H}\boldsymbol{A}_{\eta_n}\boldsymbol{\Sigma}\boldsymbol{A}_{\eta_n} \right)$$
$$+ \frac{\eta_n^2}{2} \boldsymbol{b}^\top \boldsymbol{A}_{\eta_n} \boldsymbol{H} \boldsymbol{A}_{\eta_n} \boldsymbol{b} + o(n^{-1} + \eta_n^2). \quad (7)$$

The first term in (7) is the variance component, while the second term is the teacher-induced bias. Thus, the KL term is beneficial only when the variance reduction from teacher guidance outweighs the bias introduced by teacher–target mismatch. A small-$\eta_n$ expansion makes this explicit:

$$E\{R(\widehat{\boldsymbol{\theta}}_{\eta_n})\} - R(\boldsymbol{\theta}^\star)\} - E\{R(\widehat{\boldsymbol{\theta}}_0) - R(\boldsymbol{\theta}^\star)\}$$
$$= -\frac{\eta_n}{n}\tau + \frac{\eta_n^2}{2}B + o(n^{-1} + \eta_n^2).$$

where

$$\tau = \operatorname{tr}\left( \boldsymbol{H}^{-1}\boldsymbol{G}\boldsymbol{H}^{-1}\boldsymbol{\Sigma} \right), \quad B = \boldsymbol{b}^\top \boldsymbol{H}^{-1}\boldsymbol{b}.$$

The quantity $\tau$ captures the variance-reduction direction induced by the KL term, whereas $B$ captures teacher-induced bias. At the level of this leading approximation, the KL penalty can be beneficial when $\tau > 0$ and the teacher-induced bias $B$ is moderate: the term $-(\eta_n/n)\tau$ represents variance reduction, whereas $(\eta_n^2/2)B$ represents teacher-induced bias. However, the stated remainder is not sharp enough to provide a strict improvement guarantee for weights of order $1/n$, where the variance and bias terms balance. Thus, the expansion should be interpreted as a local bias–variance explanation rather than a universal guarantee. This motivates selecting $\eta$ using internal validation data. This theoretical behavior is consistent with the empirical pattern in Sections 5.3–5.5: DiSKD benefits from

informative teachers, while validation-selected weights and robustness checks mitigate the impact of weaker or shifted teachers. The full estimator expansion and proof details are provided in Appendix A.4.

## 4. Distillation with Outcome Heterogeneity

Section 2 assumes that student and teacher share the same definition of competing events. In practice, however, external teachers may provide predictions at a coarser granularity than the internal competing-risks student. We focus on two common forms of partially aligned teacher information: (1) overall risk versus competing risks and (2) fixed-horizon binary risk versus competing risks. In both cases, the student remains a cause-specific competing-risks model. Additional collapsed single-risk variants are described in Appendix B.

### 4.1. Distill Overall Risk to Competing Risks

In real-world applications, a common heterogeneity arises when internal data record multiple competing risks, whereas teachers capture only aggregated outcomes, such as overall mortality, and do not distinguish causes. To address this, we use the fact that the overall discrete hazard is the sum of the cause-specific hazards. Let $\tilde{\lambda}(\tau_k; \boldsymbol{Z}_i)$ denote the overall-event hazard provided by the teacher. We define a time-dependent Bernoulli KL divergence:

$$d(\tilde{P}\|P_{\boldsymbol{r}}; \tau_k, \boldsymbol{Z}_i) = \tilde{\lambda}(\tau_k; \boldsymbol{Z}_i) \log\left( \frac{\tilde{\lambda}(\tau_k; \boldsymbol{Z}_i)}{\sum_{j=1}^J \lambda_j(\tau_k; \boldsymbol{Z}_i)} \right)$$
$$+ \left\{ 1 - \tilde{\lambda}(\tau_k; \boldsymbol{Z}_i) \right\} \log\left( \frac{1 - \tilde{\lambda}(\tau_k; \boldsymbol{Z}_i)}{1 - \sum_{j=1}^J \lambda_j(\tau_k; \boldsymbol{Z}_i)} \right).$$

This divergence quantifies the discrepancy between the overall-event teacher and the aggregate event probability implied by the internal competing-risks student. The accumulated KL divergence across subjects in the internal cohort is defined analogously to (3). Importantly, this term regularizes only the overall event hazard; the allocation of risk across causes is still learned from the internal competing-risks likelihood.

**Proposition 4.1.** *The penalized log-likelihood, $\ell_\eta(\boldsymbol{r}) = \ell(\boldsymbol{r}) - \eta\,\mathrm{D}(\tilde{P}\|P_{\boldsymbol{r}})$, is proportional to*

$$\ell_\eta(\boldsymbol{r}) \propto \sum_{i=1}^n \sum_{k=1}^K Y_{ik}\left[ \sum_{j=1}^J \delta_{ik}^{(j)} r_j(\tau_k; \boldsymbol{Z}_i) \right.$$
$$+ \eta\,\tilde{\lambda}(\tau_k; \boldsymbol{Z}_i) \log\left( \sum_{j=1}^J \exp\{r_j(\tau_k; \boldsymbol{Z}_i)\} \right)$$
$$\left. - (1+\eta) \log\left( 1 + \sum_{j=1}^J \exp\{r_j(\tau_k; \boldsymbol{Z}_i)\} \right) \right].$$

This formulation provides a principled way to borrow information from an overall-risk teacher while preserving the granularity of the internal competing-risks student. Conversely, if the teacher is a competing-risks model but the student target is an overall event, the same construction can be applied by first collapsing the teacher's cause-specific hazards into an overall hazard.

### 4.2. Distill Binary Outcome Model to Competing Risks

Another common form of partial alignment arises when the teacher provides only binary predictions over a fixed time window, for example whether the primary event occurs by $\tau_K$, without modeling event timing or competing risks. Let event 1 denote the primary event. Such a prediction corresponds to a horizon-specific cumulative incidence,

$$\tilde{F}_1(\tau_K; \boldsymbol{Z}_i) = \sum_{l=1}^{K} \tilde{P}(D_i = \tau_l, \Delta_i = 1 \mid \boldsymbol{Z}_i).$$

Because the student's cause-specific hazards determine the CIF through (6), we can align the teacher's binary prediction with the student's CIF prediction. Specifically, we define a Bernoulli KL divergence:

$$d(\tilde{P}\|P_{\boldsymbol{r}}; \tau_K, \boldsymbol{Z}_i) = \tilde{F}_1(\tau_K; \boldsymbol{Z}_i) \log\left(\frac{\tilde{F}_1(\tau_K; \boldsymbol{Z}_i)}{\tilde{F}_1(\tau_K; \boldsymbol{Z}_i)}\right)$$
$$+ \left\{1 - \tilde{F}_1(\tau_K; \boldsymbol{Z}_i)\right\} \log\left(\frac{1 - \tilde{F}_1(\tau_K; \boldsymbol{Z}_i)}{1 - F_1(\tau_K; \boldsymbol{Z}_i)}\right).$$

This instantiation distills simplified horizon-level teacher predictions into a detailed competing-risks student without requiring structural alignment between the two sources. In effect, the distillation leverages complementary information while preserving the granularity and interpretability of the student competing-risks model.

## 5. Experiments

Our experiments examine whether and how predictive knowledge encoded in teacher survival models can be transferred to a discrete-time competing-risks student under several forms of heterogeneity. The empirical study is organized around four questions: (i) in simulations, do the expected behaviors of DiSKD emerge under partial outcome alignment, teacher misspecification, covariate shift, multiple teachers, and limited internal sample size; (ii) on benchmark survival datasets, does DiSKD deliver consistent improvements over internal-only training, and are the gains robust across neural architectures and stable under degraded teacher information; (iii) in the Scientific Registry of Transplant Recipients (SRTR), can DiSKD support model updating when the target cohort exhibits temporal and geographic distribution shifts and the historical teacher is trained under a different regime with covariate mismatch and partial

outcome-definition alignment; and (iv) in a second SRTR case study, whether the gains persist when the downstream objective is deployability rather than model capacity, by distilling an updated high-capacity teacher into a donor-only student with constrained covariates for allocation-time donor-quality assessment.

### 5.1. Metrics

For competing risks, we evaluate discrimination using the cause-specific time-dependent concordance index ($C^{td}$), computed separately for each cause as in (Antolini et al., 2005; Lee et al., 2018a), and summarize overall fit using predictive deviance (Burnham & Anderson, 2002; Tutz et al., 2016). For single-risk outcomes, we additionally report integrated Brier score (IBS) and the negative integrated binomial log-likelihood (IBLL), to capture calibration and overall predictive accuracy over time (Gerds & Schumacher, 2006; Graf et al., 1999; Kvamme et al., 2019). For interpretability, we use SHAP and summarize feature-selection stability by the Jaccard index (Lundberg & Lee, 2017; Vo et al., 2024). Definitions and implementation details are deferred to Appendix C.

### 5.2. Tuning Parameter Selection

We tuned hyperparameters via 5-fold cross-validation using Optuna with the Tree-structured Parzen Estimator (TPE) sampler (Akiba et al., 2019). The search space covered the number of layers, hidden units, dropout rate, batch size, learning rate, the distillation weight $\eta$, and, when temperature-adjusted distillation was used, the temperature $T$. Compared with grid search, TPE avoids committing to a rigid, hand-crafted grid when reasonable ranges are unclear (as is often the case for $(T, \eta)$, whose optimal values depend on the relative strength of teacher and student information). Instead, the Bayesian sampler adaptively refines toward high-performing regions based on observed validation deviance, which reduces the risk of missing good settings due to an overly coarse or mis-specified grid. Additional details are provided in Appendix D.1.

### 5.3. Simulation

To assess robustness under increasing heterogeneity, we consider scenarios with partial outcome alignment, teacher misspecification, covariate shift, and multiple teachers. We simulate two correlated competing risks. Unless otherwise specified, we draw a teacher-training set ($n = 10,000$), a small internal student-training set ($n = 500$), and an independent test set ($n = 5,000$) from the same distribution.

**Partial Outcome Alignment.** We compare an internal-only baseline with two DiSKD variants: (1) DiSKD-C, where the teacher is a competing-risks model that pro-

vides cause-specific hazard sequences for each risk; and (2) DiSKD-O, where the teacher is an overall-event model that provides only the aggregated hazard sequence.

**Teacher Misspecification.** To assess robustness to teacher quality, we consider three scenarios reflecting different quality levels: (a) Good, (b) Fair, and (c) Poor. Details are provided in Appendix D.2. As shown in Figure 1, when the teacher is of good quality, DiSKD-C attains the highest $C^{td}$ for both risks and the lowest predictive deviance, clearly outperforming DiSKD-O. As teacher quality declines, however, the performance gap between DiSKD-C and DiSKD-O diminishes across both metrics, indicating that overall hazard predictions (DiSKD-O) become nearly as informative as cause-specific predictions when external covariate information is limited. Sensitivity to the distillation weight $\eta$ was also examined in Appendix E.1.1.

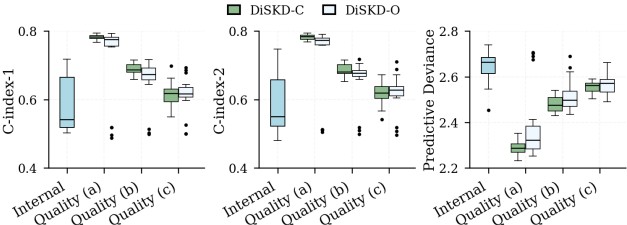

*Figure 1.* $C^{td}$ for each risk and predictive deviance on the simulated test set. Teacher quality levels (a–c) correspond to progressively reduced covariate availability in the teacher. Results were summarized over 20 random seeds.

**Covariate-Shift Robustness.** In Appendix E.1.2, we evaluate robustness to covariate distribution shift between teacher and student populations. Even under severe shift where the teacher alone may be miscalibrated on the internal test set, DiSKD remains beneficial relative to the internal-only baseline and does not exhibit negative transfer.

**Multi-Teacher Distillation.** In practice, external knowledge may come from several models whose compatibility with the internal cohort differs because of covariate availability, model quality, or population heterogeneity. DiSKD extends naturally to this setting by replacing the single-teacher KL penalty with a weighted sum of $M$ teacher-specific KL penalties,

$$\ell_{\boldsymbol{\eta}}(\boldsymbol{r}) = \ell(\boldsymbol{r}) - \sum_{m=1}^{M} \eta_m \mathrm{D}(\tilde{P}_m \| P_{\boldsymbol{r}}),$$

where $\eta_m \geq 0$ controls the contribution of teacher $m$.

We evaluate this extension in simulation using two teachers with different covariate availability, so that one teacher is substantially more informative than the other. We compare joint distillation from both teachers and single-teacher distillation from each teacher. As shown in Appendix E.1.3,

validation selects a larger weight for the stronger teacher and a smaller weight for the weaker teacher. Joint distillation improves over relying on the weaker teacher alone and remains close to the stronger-teacher solution, suggesting that DiSKD can combine multiple external prediction sources without reducing to a naive average.

**Temperature for Correlated Competing Risks.** Temperature scaling makes cross-risk information usable in the KL term: for $T > 1$, the teacher distribution is softened so that non-target risks retain non-negligible mass, allowing the student to learn relative contrasts across causes rather than an effectively one-hot signal. To assess this effect, we vary $T \in \{1, 2, 3, 4, 5\}$ under a good-quality teacher and report test-set $C^{td}$ and predictive deviance in Figure 2. At $T = 1$, DiSKD-C exhibits noticeably larger variability and weaker average performance. Moving to $T = 2$ substantially improves both stability and predictive accuracy, while results change little for $T \geq 3$, indicating a practical plateau once the guidance is sufficiently softened.

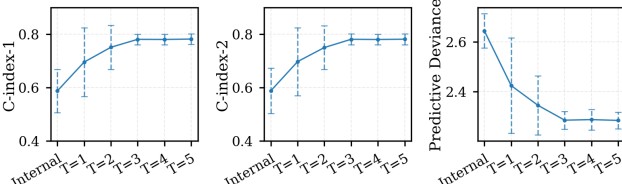

*Figure 2.* Effect of distillation temperature $T$ on the predictive performance of DiSKD-C under a high-quality teacher. $C^{td}$ for each risk and predictive deviance on the test set are reported. Results were summarized over 20 random seeds.

### 5.4. Benchmark Survival Datasets

**Datasets.** We consider three benchmark survival datasets: SUPPORT, METABRIC, and Rotterdam & GBSG (Knaus et al., 1995; Curtis et al., 2012; Foekens et al., 2000; Schumacher et al., 1994). Dataset descriptions and preprocessing details are provided in Appendix D.3.

**Consistent Gains Across Datasets and Architectures.** We consider a held-out test set (20%), a small internal student cohort (4%), and a larger teacher cohort (76%). On SUPPORT, we evaluate DiSKD across multiple discrete-time architectures and systematically degrade the teacher by restricting its covariate set, inducing a controlled loss of informativeness. On METABRIC and Rotterdam & GBSG, we fix a strong backbone (Transformer) and use all covariates to focus on cross-dataset generalization. Overall, DiSKD improves over internal-only training across datasets and architectures, and remains robust as the teacher is degraded, suggesting a general mechanism for leveraging teacher survival predictions across cohorts with differing event profiles and covariate structures (Appendix E.2.2).

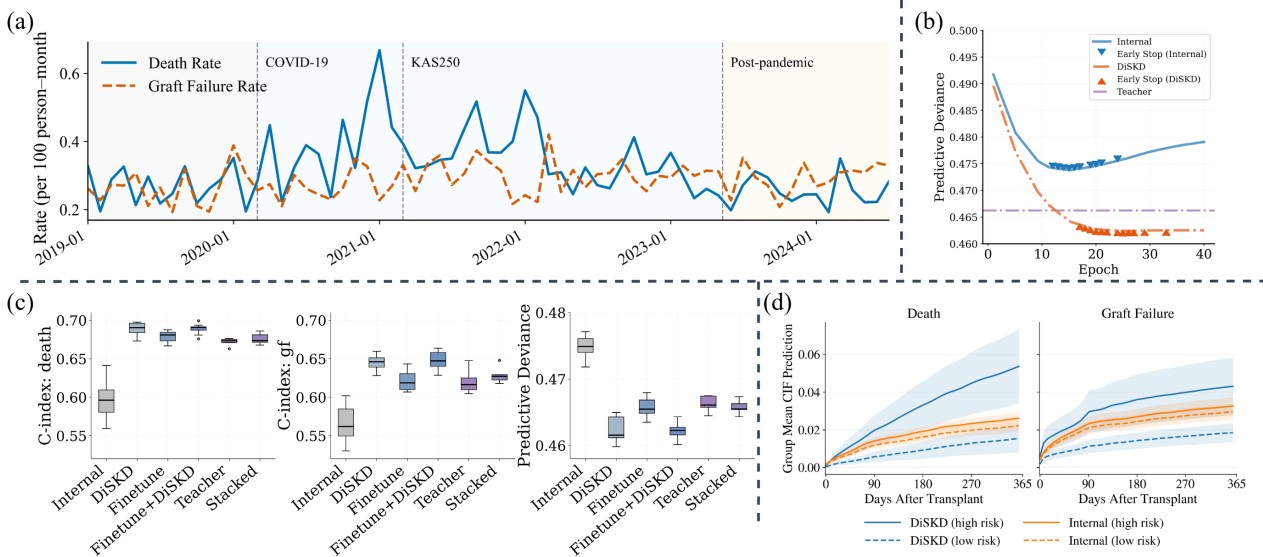

*Figure 3.* Evidence of distribution shift and benefits of DiSKD. (a) temporal changes in event rates across COVID and policy eras; (b) training dynamics under early stopping; (c) test performance across strategies; (d) risk-stratified CIF trajectories for death and graft failure.

## 5.5. SRTR Model Updating Under Population Shifts

We study SRTR model updating across pandemic- and policy-driven regime change, transferring one-year death and graft-failure risk from a large, shifted COVID+KAS250 cohort (teacher) to a post-COVID target cohort (student), see Appendix D.4. This setting features practical mismatches: temporal/geospatial drift with event-rate shift (often yielding miscalibration; Figure 3a; Appendix E.3.1), limited access to the teacher (predictions only), different covariate availability, and partially aligned outcomes (e.g. aggregated or single-cause guidance). DiSKD is designed to accommodate such heterogeneous teacher information.

**Evaluation Protocol.** We discretize follow-up into weekly intervals and fit discrete-time competing-risks models using donor/recipient covariates and transplant-center indicators. Methods are evaluated on the post-COVID cohort using 10 repeats of 5-fold cross-validation. We compare six strategies: (1) Internal (post-COVID only), (2) Teacher (COVID+KAS250 applied to post-COVID), (3) Stacked (pooled training), (4) Finetune (initialized from the teacher and finetuned on post-COVID), (5) Finetune+DiSKD, and (6) DiSKD (post-COVID with KL guidance).

**Stabilized Updating Dynamics.** On the post-COVID cohort, the internal-only model overfits quickly: validation deviance improves briefly then deteriorates. In contrast, DiSKD improves steadily and converges to lower deviance, with later stopping points (Figure 3b). The teacher is competitive but plateaus above DiSKD, reflecting residual COVID-to-post-COVID shift. Overall, DiSKD stabilizes

updating and improves accuracy while reducing overfitting.

**Robustness to Heterogeneity.** DiSKD provides the best concordance–deviance trade-off, outperforming the internal baseline, COVID-era teacher, and stacking (Figure 3c). Stacking yields modest, less reliable gains, and finetuning improves over the teacher but remains below DiSKD; Finetune+DiSKD is comparable to DiSKD, suggesting the KL term drives the benefit. DiSKD remains robust when the teacher architecture varies and covariates are degraded, typically outperforming internal-only and direct teacher deployment (Appendix E.3.2).

**Robustness to Outcome Granularity.** Beyond distributional and covariate shift, external models may also report a coarser outcome object than the target competing-risks analysis. We therefore test the partial-alignment formulations in Section 4.1 and Section 4.2 by training COVID+KAS250 teachers with three output granularities: matched competing risks, overall event risk, and fixed-horizon binary event-specific risk. Each teacher is distilled into the same post-COVID competing-risks student.

As shown in Table 1, all teacher granularities improve over the internal-only competing-risks model. The matched competing-risks teacher gives the lowest predictive deviance, while the overall-event teacher achieves comparable discrimination and the best graft-failure concordance, showing that aggregated event information can still provide useful regularization for an event-sparse target cohort. The binary teachers are weaker, as expected because they discard timing and competing-event information, but they remain

beneficial over internal-only training. These results support the main modeling claim: DiSKD does not require a perfectly matched external competing-risks teacher, but can use partially aligned teacher outputs while preserving the competing-risks target of the student.

*Table 1.* Performance on the post-COVID SRTR target cohort under different COVID+KAS250 teacher outcome granularities. All teachers are distilled into the same competing-risks student. CR denotes competing risks; the overall-event teacher predicts any event, and binary event-$k$ predicts event $k$ versus not experiencing event $k$ by that horizon. Entries are mean (standard deviation).

| Method | $C^{td}$ (Death) | $C^{td}$ (Graft) | Pred Dev. |
|---|---|---|---|
| Internal CR | 0.5954 (0.0232) | 0.5667 (0.0225) | 0.4749 (0.0015) |
| CR → CR | 0.6889 (0.0078) | 0.6451 (0.0094) | 0.4624 (0.0018) |
| Overall → CR | 0.6887 (0.0049) | 0.6533 (0.0064) | 0.4650 (0.0013) |
| Binary-1 → CR | 0.6668 (0.0115) | 0.5967 (0.0136) | 0.4702 (0.0019) |
| Binary-2 → CR | 0.6306 (0.0060) | 0.6453 (0.0039) | 0.4708 (0.0020) |

**Risk Stratification.** Using predicted one-year CIFs on held-out post-COVID patients, we split by the median into high/low-risk groups and plot mean CIF trajectories (Figure 3d). For both death and graft failure, DiSKD yields a larger, more persistent separation than internal-only training, indicating improved discrimination on an absolute-risk scale relevant to allocation-time decisions.

### 5.6. SRTR Donor-Quality Prediction

Kidney donor allocation decisions require accurate donor-only risk prediction computed immediately at organ offer. We therefore examine whether an updated donor+recipient+center model (strengthened via DiSKD) can be distilled into a deployable donor-only tool aligned with Kidney Donor Risk Index (KDRI). Using the post-COVID competing-risks model as the teacher, we distill cause-specific hazards for death and graft failure into a lightweight donor-only student under two feature sets: all donor variables and a KDRI-compatible subset (eight donor factors). We compare against a donor-only model without distillation and against KDRI; full details are in Appendix D.5.

Figure 4 summarizes post-COVID cause-specific discrimination. The donor-only baseline trained on post-COVID data is close to KDRI, indicating limited gains from refitting alone. In contrast, the distilled donor-only student achieves higher $C^{td}$ for death and graft failure, showing that DiSKD transfers signal from a richer donor+recipient+center teacher into a deployable donor-only predictor. These gains persist under KDRI-compatible inputs: with only the eight KDRI donor factors (Figure 4b), distillation improves over the donor-only baseline and exceeds KDRI. Operationally, DiSKD compresses a competing-risks model into a donor-only tool while preserving the familiar KDRI-style interface for organ-offer decisions.

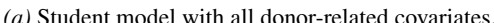
*(a)* Student model with all donor-related covariates.

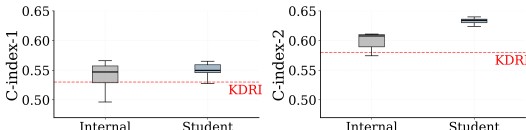

*(b)* Student model with eight KDRI donor factors.

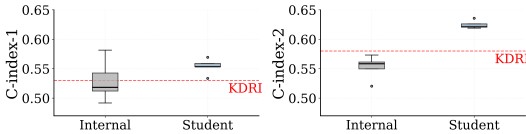

*Figure 4.* Discrimination comparison of donor-only student models on the post-COVID test cohort. The student is a donor-only model distilled from the all-covariates teacher. Panel (a) uses all donor-related covariates available in SRTR; Panel (b) restricts inputs to the eight donor factors used in the current KDRI formulation. The red dashed line indicates the performance of the KDRI benchmark.

## 6. Discussion

We proposed a KL-based knowledge distillation framework for discrete-time competing risks that aligns student hazards with teacher predictions, offering a principled alternative to pooling or finetuning in data-limited, heterogeneous, and privacy-constrained settings. Simulations showed improved discrimination and calibration, robust even under degraded teacher inputs, and SRTR applications demonstrated consistent gains over internal-only, teacher-only, stacked, and finetuning strategies despite a substantial temporal shift. We further showed that DiSKD can use partially aligned teacher outputs, including overall-risk and fixed-horizon binary predictions, while preserving the competing-risks target of the student model.

The proposed framework can be extended in several directions beyond the discrete-time competing-risks setting studied here. First, KL-based distillation can be adapted to continuous-time survival models, including Cox-type models, where alignment must be defined through cumulative hazards, risk sets, or partial-likelihood components. Second, related distillation ideas may be developed for composite-likelihood objectives based on pairwise comparability or risk-set contributions. These extensions require likelihood-specific KL constructions that are distinct from the interval-level multinomial formulation used in DiSKD, and we will report them in separate works.

Overall, DiSKD provides a flexible and privacy-conscious distillation framework for knowledge transfer in evolving clinical environments. By operating at the level of predicted hazards rather than raw data or shared parameters, it supports transfer across heterogeneous populations, feature sets, and outcome definitions while retaining the discrete-time competing-risks structure needed for cause-specific risk prediction.

## Acknowledgements

The data reported here have been supplied by the Hennepin Healthcare Research Institute (HHRI) as the contractor for the Scientific Registry of Transplant Recipients (SRTR). The interpretation and reporting of these data are the responsibility of the authors and should not be construed as an official policy or interpretation of the SRTR or the U.S. Government. This work was partially supported by the National Institutes of Health grant DK129539.

## Impact Statement

This paper advances survival analysis and machine learning by developing methods for discrete-time competing risks that improve prediction and inference in data-limited, heterogeneous settings. Although motivated by organ transplantation, the approach applies broadly to time-to-event problems with multiple event types, such as oncology (cancer-related death vs other death), cardiovascular outcomes, and health services research. By enabling privacy-conscious transfer of predictive structure from external models without sharing individual-level data, and by providing mechanisms to mitigate negative transfer under outcome-definition mismatch, this work can help extend high-quality risk prediction to underrepresented cohorts and rare-event clinical settings where large, fully harmonized datasets are difficult to obtain.

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

# A. Proofs

## A.1. Proof of Proposition 2.1

*Proof.* Let $\mathcal{S} = \{(i,k) : Y_{ik} = 1\}$ be the set of at-risk subject–interval pairs and $N = |\mathcal{S}| = \sum_{i=1}^{n} \sum_{k=1}^{K} Y_{ik}$. At each $(i,k) \in \mathcal{S}$, define a $(J+1)$-category outcome $U_{ik} \in \{0, 1, \dots, J\}$, where $U_{ik} = j$ indicates an event of cause $j$ at $\tau_k$ and $U_{ik} = 0$ indicates "no event" at $\tau_k$. The model-implied conditional distribution at $(i,k)$ is

$$p_{ik}^{(j)}(\boldsymbol{r}) := P_{\boldsymbol{r}}(U_{ik} = j \mid Y_{ik} = 1, \boldsymbol{Z}_i) = \lambda_j(\tau_k; \boldsymbol{Z}_i), \quad j = 1, \dots, J,$$

and

$$p_{ik}^{(0)}(\boldsymbol{r}) := P_{\boldsymbol{r}}(U_{ik} = 0 \mid Y_{ik} = 1, \boldsymbol{Z}_i) = 1 - \lambda(\tau_k; \boldsymbol{Z}_i), \quad \lambda(\tau_k; \boldsymbol{Z}_i) = \sum_{j=1}^{J} \lambda_j(\tau_k; \boldsymbol{Z}_i).$$

Similarly, the empirical conditional distribution at $(i,k)$ is the one-hot vector $q_{ik}$ induced by the observed indicators:

$$q_{ik}^{(j)} := \delta_{ik}^{(j)}, \quad j = 1, \dots, J, \quad q_{ik}^{(0)} := 1 - \sum_{j=1}^{J} \delta_{ik}^{(j)}.$$

We define the empirical KL divergence between the collection of empirical conditionals $\{q_{ik}\}_{(i,k)\in\mathcal{S}}$ and the model conditionals $\{p_{ik}(\boldsymbol{r})\}_{(i,k)\in\mathcal{S}}$ as

$$\mathrm{D}(P_n\|P_{\boldsymbol{r}}) := \sum_{(i,k)\in\mathcal{S}} \mathrm{KL}(q_{ik} \| p_{ik}(\boldsymbol{r})) = \sum_{i=1}^{n} \sum_{k=1}^{K} Y_{ik} \sum_{c=0}^{J} q_{ik}^{(c)} \log \frac{q_{ik}^{(c)}}{p_{ik}^{(c)}(\boldsymbol{r})}. \tag{8}$$

Because $q_{ik}$ is one-hot, $\sum_{c=0}^{J} q_{ik}^{(c)} \log q_{ik}^{(c)} = 0$ (with the convention $0 \log 0 = 0$), and hence

$$\mathrm{KL}(q_{ik}\|p_{ik}(\boldsymbol{r})) = - \sum_{c=0}^{J} q_{ik}^{(c)} \log p_{ik}^{(c)}(\boldsymbol{r}).$$

Substituting the definitions of $q_{ik}$ and $p_{ik}(\boldsymbol{r})$ yields

$$-\mathrm{D}(P_n\|P_{\boldsymbol{r}}) = \sum_{i=1}^{n} \sum_{k=1}^{K} Y_{ik} \left[ \sum_{j=1}^{J} \delta_{ik}^{(j)} \log \lambda_j(\tau_k; \boldsymbol{Z}_i) + \left(1 - \sum_{j=1}^{J} \delta_{ik}^{(j)}\right) \log\{1 - \lambda(\tau_k; \boldsymbol{Z}_i)\} \right]. \tag{9}$$

The right-hand side is exactly the internal discrete-time competing risks log-likelihood expressed in the $(J+1)$-multinomial form. Therefore,

$$\ell(\boldsymbol{r}) = -\mathrm{D}(P_n\|P_{\boldsymbol{r}}), \tag{10}$$

up to an irrelevant constant/scale depending on whether one uses the summed or averaged version of (8).

Next, by definition of KL divergence, the accumulated divergence to the teacher conditional distribution collection $\{\tilde{p}_{ik}\}$ satisfies

$$\mathrm{D}(\tilde{P}\|P_{\boldsymbol{r}}) = \sum_{(i,k)\in\mathcal{S}} \mathrm{KL}(\tilde{p}_{ik} \| p_{ik}(\boldsymbol{r})) = - \sum_{(i,k)\in\mathcal{S}} \sum_{c=0}^{J} \tilde{p}_{ik}^{(c)} \log p_{ik}^{(c)}(\boldsymbol{r}) + \mathrm{const},$$

where $\mathrm{const}$ does not depend on $\boldsymbol{r}$.

Combining the two displays, the penalized objective in (4) can be written as

$$\ell_\eta(\boldsymbol{r}) = \ell(\boldsymbol{r}) - \eta \, \mathrm{D}(\tilde{P}\|P_{\boldsymbol{r}}) \propto -\mathrm{D}(P_n\|P_{\boldsymbol{r}}) - \eta \, \mathrm{D}(\tilde{P}\|P_{\boldsymbol{r}}).$$

Multiplying by the positive constant $\frac{1}{1+\eta}$ does not change the optimizer. Let $\alpha = \eta/(1+\eta)$. Then maximizing $\ell_\eta(\boldsymbol{r})$ is equivalent to minimizing

$$(1-\alpha) \, \mathrm{D}(P_n\|P_{\boldsymbol{r}}) + \alpha \, \mathrm{D}(\tilde{P}\|P_{\boldsymbol{r}}),$$

which proves the claim. $\qquad\square$

### A.2. Proof of Proposition 2.2

*Proof.* Start from $\ell_\eta(\boldsymbol{r}) = \ell(\boldsymbol{r}) - \eta\, D(\tilde{P}\|P_{\boldsymbol{r}})$. For each $(i,k)$ with $Y_{ik} = 1$, the time-dependent KL divergence is

$$d(\tilde{P}\|P_{\boldsymbol{r}}; \boldsymbol{Z}_i, \tau_k) = \sum_{j=1}^J \tilde{\lambda}_j \log \frac{\tilde{\lambda}_j}{\lambda_j} + (1 - \tilde{\lambda})\log \frac{1 - \tilde{\lambda}}{1 - \lambda},$$

where we suppress $(\tau_k; \boldsymbol{Z}_i)$ in notation for brevity.

Expanding and dropping terms independent of $\boldsymbol{r}$ yields

$$-\eta\, d(\tilde{P}\|P_{\boldsymbol{r}}; \boldsymbol{Z}_i, \tau_k) \propto \eta \left[ \sum_{j=1}^J \tilde{\lambda}_j \log \lambda_j + (1 - \tilde{\lambda})\log(1 - \lambda) \right].$$

Combining with the internal contribution at $(i,k)$,

$$\ell(\boldsymbol{r}) = \sum_{i,k} Y_{ik} \left[ \sum_{j=1}^J \delta_{ik}^{(j)} \log \lambda_j + \left(1 - \sum_{j=1}^J \delta_{ik}^{(j)}\right) \log(1 - \lambda) \right],$$

we obtain, up to additive constants,

$$\ell_\eta(\boldsymbol{r}) \propto \sum_{i,k} Y_{ik} \left[ \sum_{j=1}^J \{\delta_{ik}^{(j)} + \eta\tilde{\lambda}_j\} \log \lambda_j + \left\{1 - \sum_{j=1}^J \delta_{ik}^{(j)} + \eta(1 - \tilde{\lambda})\right\} \log(1 - \lambda) \right].$$

Now substitute:

$$\log \lambda_j = r_j - \log\left(1 + \sum_{j'} e^{r_{j'}}\right), \qquad \log(1 - \lambda) = -\log\left(1 + \sum_{j'} e^{r_{j'}}\right).$$

The coefficient in front of the shared term $-\log\left(1 + \sum_{j'} e^{r_{j'}}\right)$ becomes

$$\sum_{j=1}^J \{\delta_{ik}^{(j)} + \eta\tilde{\lambda}_j\} + \left\{1 - \sum_{j=1}^J \delta_{ik}^{(j)} + \eta(1 - \tilde{\lambda})\right\} = 1 + \eta.$$

Therefore,

$$\ell_\eta(\boldsymbol{r}) \propto \sum_{i,k} Y_{ik} \left[ \sum_{j=1}^J \{\delta_{ik}^{(j)} + \eta\tilde{\lambda}_j\} r_j - (1 + \eta)\log\left(1 + \sum_{j=1}^J e^{r_j}\right) \right].$$

Dividing by $(1 + \eta)$ (a positive constant) does not change the maximizer, yielding

$$\ell_\eta(\boldsymbol{r}) \propto \sum_{i,k} Y_{ik} \left[ \sum_{j=1}^J \frac{\delta_{ik}^{(j)} + \eta\tilde{\lambda}_j}{1 + \eta} r_j - \log\left(1 + \sum_{j=1}^J e^{r_j}\right) \right],$$

which is exactly (5). $\qquad\square$

### A.3. Proof of Proposition 4.1

*Proof.* As in Appendix A.2, expand the Bernoulli KL and drop terms independent of $\boldsymbol{r}$:

$$-\eta\, d(\tilde{P}\|P_{\boldsymbol{r}}) \propto \eta \left[ \tilde{\lambda} \log \lambda + (1 - \tilde{\lambda})\log(1 - \lambda) \right].$$

Add the internal multinomial log-likelihood written in terms of $\log \lambda_j$ and $\log(1 - \lambda)$. Using $\log \lambda = \log\{\sum_j \lambda_j\}$ and the softmax form,

$$\lambda = \frac{\sum_{j=1}^J e^{r_j}}{1 + \sum_{j=1}^J e^{r_j}} \quad \Rightarrow \quad \log \lambda = \log\left(\sum_{j=1}^J e^{r_j}\right) - \log\left(1 + \sum_{j=1}^J e^{r_j}\right),$$

and $\log(1 - \lambda) = -\log\left(1 + \sum_j e^{r_j}\right)$. Collecting terms yields the stated expression. $\qquad\square$

## A.4. Theoretical Details for the Local Excess-Risk Expansion

This appendix provides the full assumptions and proof for the local excess-risk expansion in Section 3. The result is intended to characterize the local bias–variance role of the KL penalty, rather than to provide a universal improvement guarantee.

For notational simplicity, we present the result for a discrete-time survival model with one event type. The competing-risks case follows by replacing the Bernoulli interval-level outcome with the corresponding multinomial interval-level outcome distribution over no event and the $J$ competing causes.

### A.4.1. SETUP AND NOTATION

Let $\boldsymbol{\theta} \in \Theta \subset \mathbb{R}^p$ denote the parameters of a differentiable discrete-time hazard model. For subject $i$, let $\boldsymbol{Z}_i$ be covariates, let $\tau_1, \ldots, \tau_K$ denote the discrete time grid, let $Y_{ik}$ be the at-risk indicator, and let $\delta_{ik}$ indicate an event at $\tau_k$. The student model specifies the conditional hazard

$$\lambda_{\boldsymbol{\theta}}(\tau_k; \boldsymbol{Z}_i) = \sigma\{r_{\boldsymbol{\theta}}(\tau_k; \boldsymbol{Z}_i)\},$$

where $\sigma(u) = \{1 + \exp(-u)\}^{-1}$. The internal log-likelihood contribution for subject $i$ is

$$\ell_i(\boldsymbol{\theta}) = \sum_{k=1}^{K} Y_{ik} \left[ \delta_{ik} \log \lambda_{\boldsymbol{\theta}}(\tau_k; \boldsymbol{Z}_i) + (1 - \delta_{ik}) \log\{1 - \lambda_{\boldsymbol{\theta}}(\tau_k; \boldsymbol{Z}_i)\} \right].$$

Suppose the teacher provides hazards $\tilde{\lambda}(\tau_k; \boldsymbol{Z}_i)$ on the same time grid. The accumulated teacher-student KL contribution for subject $i$ is

$$q_i(\boldsymbol{\theta}) = \sum_{k=1}^{K} Y_{ik} \left[ \tilde{\lambda}(\tau_k; \boldsymbol{Z}_i) \log \frac{\tilde{\lambda}(\tau_k; \boldsymbol{Z}_i)}{\lambda_{\boldsymbol{\theta}}(\tau_k; \boldsymbol{Z}_i)} + \{1 - \tilde{\lambda}(\tau_k; \boldsymbol{Z}_i)\} \log \frac{1 - \tilde{\lambda}(\tau_k; \boldsymbol{Z}_i)}{1 - \lambda_{\boldsymbol{\theta}}(\tau_k; \boldsymbol{Z}_i)} \right].$$

Define the empirical internal risk, empirical KL criterion, and KL-integrated objective as

$$R_n(\boldsymbol{\theta}) = -\frac{1}{n} \sum_{i=1}^{n} \ell_i(\boldsymbol{\theta}), \qquad Q_n(\boldsymbol{\theta}) = \frac{1}{n} \sum_{i=1}^{n} q_i(\boldsymbol{\theta}), \qquad M_{n,\eta}(\boldsymbol{\theta}) = R_n(\boldsymbol{\theta}) + \eta Q_n(\boldsymbol{\theta}).$$

Let

$$R(\boldsymbol{\theta}) = E\{R_n(\boldsymbol{\theta})\}, \qquad Q(\boldsymbol{\theta}) = E\{Q_n(\boldsymbol{\theta})\}, \qquad M_\eta(\boldsymbol{\theta}) = R(\boldsymbol{\theta}) + \eta Q(\boldsymbol{\theta}).$$

Let $\boldsymbol{\theta}^\star$ be the target parameter minimizing the internal population risk $R(\boldsymbol{\theta})$. Define

$$\boldsymbol{H} = \nabla^2 R(\boldsymbol{\theta}^\star), \qquad \boldsymbol{b} = \nabla Q(\boldsymbol{\theta}^\star), \qquad \boldsymbol{G} = \nabla^2 Q(\boldsymbol{\theta}^\star).$$

Here $\boldsymbol{b}$ measures the local teacher-target discrepancy: when the teacher hazards align with the target population, $\boldsymbol{b}$ is small or zero; when the teacher is mismatched, $\boldsymbol{b}$ captures the local direction in which the KL penalty pulls the student away from the internal target.

### A.4.2. LOCAL REGULARITY ASSUMPTIONS

**Assumption A.1** (Local regularity). There exists a neighborhood $\mathcal{N}$ of $\boldsymbol{\theta}^\star$ such that the following conditions hold.

(B1) **Interior target and local identifiability.** The parameter $\boldsymbol{\theta}^\star$ lies in the interior of $\Theta$ and is the unique minimizer of $R(\boldsymbol{\theta})$ in $\mathcal{N}$.

(B2) **Smoothness.** The functions $R(\boldsymbol{\theta})$ and $Q(\boldsymbol{\theta})$ are three times continuously differentiable in $\mathcal{N}$. The empirical criteria $R_n(\boldsymbol{\theta})$ and $Q_n(\boldsymbol{\theta})$ are twice continuously differentiable in $\mathcal{N}$ with probability tending to one.

(B3) **Local curvature.** The Hessian $\boldsymbol{H} = \nabla^2 R(\boldsymbol{\theta}^\star)$ is positive definite. Moreover, there exists $\bar{\eta} > 0$ such that $\boldsymbol{H} + \eta \boldsymbol{G}$ is nonsingular for all $\eta \in [0, \bar{\eta}]$.

(B4) **Score behavior.** The internal score satisfies

$$\sqrt{n}\,\nabla R_n(\boldsymbol{\theta}^\star) = \boldsymbol{\xi}_n, \qquad E(\boldsymbol{\xi}_n) = 0, \qquad E(\boldsymbol{\xi}_n \boldsymbol{\xi}_n^\top) \to \boldsymbol{\Sigma},$$

for some finite positive semidefinite matrix $\boldsymbol{\Sigma}$.

(B5) **Uniform Hessian convergence.** Uniformly over $\boldsymbol{\theta} \in \mathcal{N}$,

$$\nabla^2 R_n(\boldsymbol{\theta}) = \nabla^2 R(\boldsymbol{\theta}) + o_p(1), \qquad \nabla^2 Q_n(\boldsymbol{\theta}) = \nabla^2 Q(\boldsymbol{\theta}) + o_p(1).$$

(B6) **KL-score concentration.** The KL score satisfies

$$\nabla Q_n(\boldsymbol{\theta}^\star) = \nabla Q(\boldsymbol{\theta}^\star) + o_p(1) = \boldsymbol{b} + o_p(1).$$

(B7) **Local empirical minimizer.** For any deterministic sequence $\eta_n \to 0$ with $0 \leq \eta_n \leq \bar{\eta}$, there exists a local empirical minimizer $\widehat{\boldsymbol{\theta}}_{\eta_n}$ of $M_{n,\eta_n}(\boldsymbol{\theta})$ in $\mathcal{N}$ such that

$$\widehat{\boldsymbol{\theta}}_{\eta_n} \to_p \boldsymbol{\theta}^\star.$$

(B8) **Second-order expansion for risk comparison.** The third-order remainder in the Taylor expansion of $R(\boldsymbol{\theta})$ around $\boldsymbol{\theta}^\star$ is negligible in expectation at the scale considered below. In particular, for

$$a_n = n^{-1} + \eta_n^2,$$

the estimator satisfies

$$E\left[\|\widehat{\boldsymbol{\theta}}_{\eta_n} - \boldsymbol{\theta}^\star\|^3\right] = o(a_n).$$

*Remark* A.2. Assumption A.1 places the KL-integrated deep survival model in a regular local $M$-estimation regime around $\boldsymbol{\theta}^\star$. The curvature condition is local and does not assert global convexity of the neural-network objective. Conditions (B1)–(B7) are sufficient for the estimator linearization and the leading excess-risk expansion. Condition (B8) is used only for the second-order excess-risk expansion.

A.4.3. FULL LOCAL EXPANSION

**Theorem A.3** (Full local expansion). *Assume Assumption A.1. For $\eta \in [0, \bar{\eta}]$, define*

$$\boldsymbol{A}_\eta = (\boldsymbol{H} + \eta \boldsymbol{G})^{-1}.$$

*Let $\eta_n \to 0$ be deterministic, $0 \leq \eta_n \leq \bar{\eta}$, and define*

$$a_n = n^{-1} + \eta_n^2.$$

*Then*

$$\widehat{\boldsymbol{\theta}}_{\eta_n} - \boldsymbol{\theta}^\star = -\boldsymbol{A}_{\eta_n}\left\{n^{-1/2}\boldsymbol{\xi}_n + \eta_n \boldsymbol{b}\right\} + \boldsymbol{s}_n, \qquad E\|\boldsymbol{s}_n\|^2 = o(a_n).$$

*Consequently,*

$$E\left[R(\widehat{\boldsymbol{\theta}}_{\eta_n}) - R(\boldsymbol{\theta}^\star)\right] = \frac{1}{2n}\,\mathrm{tr}\left(\boldsymbol{H}\boldsymbol{A}_{\eta_n}\boldsymbol{\Sigma}\boldsymbol{A}_{\eta_n}\right)$$
$$+ \frac{\eta_n^2}{2}\boldsymbol{b}^\top \boldsymbol{A}_{\eta_n}\boldsymbol{H}\boldsymbol{A}_{\eta_n}\boldsymbol{b} + o(a_n). \tag{11}$$

*Moreover, as $\eta \downarrow 0$,*

$$\boldsymbol{A}_\eta = \boldsymbol{H}^{-1} - \eta \boldsymbol{H}^{-1}\boldsymbol{G}\boldsymbol{H}^{-1} + O(\eta^2),$$

*and therefore*

$$E\left[R(\widehat{\boldsymbol{\theta}}_{\eta_n}) - R(\boldsymbol{\theta}^\star)\right] = \frac{1}{2n}\,\mathrm{tr}\left(\boldsymbol{H}^{-1}\boldsymbol{\Sigma}\right) - \frac{\eta_n}{n}\tau + \frac{\eta_n^2}{2}B + o(n^{-1} + \eta_n^2), \tag{12}$$

*where*

$$\tau = \mathrm{tr}\left(\boldsymbol{H}^{-1}\boldsymbol{G}\boldsymbol{H}^{-1}\boldsymbol{\Sigma}\right), \qquad B = \boldsymbol{b}^\top \boldsymbol{H}^{-1}\boldsymbol{b}.$$

The leading nonconstant terms in (12) identify the local bias–variance trade-off induced by the KL penalty. The term $-(\eta_n/n)\tau$ is the leading variance-reduction contribution, whereas $(\eta_n^2/2)B$ is the teacher-induced bias contribution. Because the remainder in (12) is only $o(n^{-1} + \eta_n^2)$, this expansion does not by itself determine the sign of the risk difference when $\eta_n = O(n^{-1})$, the scale at which the leading variance and bias terms balance. A strict improvement guarantee at that scale would require sharper second-order remainder control.

*Remark* A.4. The expansion above explains why teacher guidance may help in small or event-sparse internal cohorts: a small KL penalty can reduce variance, but a mismatched teacher induces bias. The theorem does not establish a universal strict-improvement guarantee. In particular, for $\eta_n = c/n$, the leading variance and bias terms are both of order $n^{-2}$, whereas the stated remainder is not sharp enough to determine the sign of the risk difference. This is why $\eta$ is selected by internal validation in our experiments.

### A.4.4. PROOFS

*Proof of Theorem A.3.* Let

$$\boldsymbol{\delta}_n = \widehat{\boldsymbol{\theta}}_{\eta_n} - \boldsymbol{\theta}^\star.$$

**Step 1: Local score expansion.** Because $\widehat{\boldsymbol{\theta}}_{\eta_n}$ is a local empirical minimizer, the first-order condition gives

$$\nabla M_{n,\eta_n}(\widehat{\boldsymbol{\theta}}_{\eta_n}) = \mathbf{0}.$$

Expanding around $\boldsymbol{\theta}^\star$ yields

$$\mathbf{0} = \nabla R_n(\boldsymbol{\theta}^\star) + \eta_n \nabla Q_n(\boldsymbol{\theta}^\star) + \left\{ \nabla^2 R_n(\boldsymbol{\theta}^\star) + \eta_n \nabla^2 Q_n(\boldsymbol{\theta}^\star) \right\} \boldsymbol{\delta}_n + \boldsymbol{\rho}_n,$$

where the Taylor remainder $\boldsymbol{\rho}_n$ satisfies

$$\|\boldsymbol{\rho}_n\| = o_p(\|\boldsymbol{\delta}_n\|)$$

under the smoothness and local convergence assumptions.

By Assumption A.1,

$$\nabla R_n(\boldsymbol{\theta}^\star) = n^{-1/2}\boldsymbol{\xi}_n, \qquad \nabla Q_n(\boldsymbol{\theta}^\star) = \boldsymbol{b} + o_p(1),$$

and

$$\nabla^2 R_n(\boldsymbol{\theta}^\star) + \eta_n \nabla^2 Q_n(\boldsymbol{\theta}^\star) = \boldsymbol{H} + \eta_n \boldsymbol{G} + o_p(1).$$

Therefore,

$$\mathbf{0} = n^{-1/2}\boldsymbol{\xi}_n + \eta_n \boldsymbol{b} + (\boldsymbol{H} + \eta_n \boldsymbol{G})\boldsymbol{\delta}_n + \boldsymbol{r}_n,$$

where the remainder $\boldsymbol{r}_n$ satisfies

$$E\|\boldsymbol{r}_n\|^2 = o(n^{-1} + \eta_n^2).$$

**Step 2: Estimator linearization.** Since $\boldsymbol{H} + \eta_n \boldsymbol{G}$ is nonsingular for all sufficiently large $n$, multiplying the preceding display by

$$\boldsymbol{A}_{\eta_n} = (\boldsymbol{H} + \eta_n \boldsymbol{G})^{-1}$$

gives

$$\boldsymbol{\delta}_n = -\boldsymbol{A}_{\eta_n}\left\{ n^{-1/2}\boldsymbol{\xi}_n + \eta_n \boldsymbol{b} \right\} + \boldsymbol{s}_n,$$

with

$$\boldsymbol{s}_n = -\boldsymbol{A}_{\eta_n}\boldsymbol{r}_n.$$

Because $\boldsymbol{A}_\eta$ is uniformly bounded for $\eta \in [0, \bar{\eta}]$, it follows that

$$E\|\boldsymbol{s}_n\|^2 = o(n^{-1} + \eta_n^2) = o(a_n).$$

**Step 3: Excess-risk expansion.** Since $\boldsymbol{\theta}^\star$ minimizes $R(\boldsymbol{\theta})$, we have

$$\nabla R(\boldsymbol{\theta}^\star) = \mathbf{0}.$$

A second-order Taylor expansion gives

$$R(\widehat{\boldsymbol{\theta}}_{\eta_n}) - R(\boldsymbol{\theta}^\star) = \frac{1}{2}\boldsymbol{\delta}_n^\top \boldsymbol{H}\boldsymbol{\delta}_n + o(\|\boldsymbol{\delta}_n\|^2).$$

Using the linearization from Step 2 and Assumption A.1, the contribution of $\boldsymbol{s}_n$ is $o(a_n)$ in expectation. Thus,

$$E\left[R(\widehat{\boldsymbol{\theta}}_{\eta_n}) - R(\boldsymbol{\theta}^\star)\right] = \frac{1}{2}E\left[\left\{n^{-1/2}\boldsymbol{\xi}_n + \eta_n\boldsymbol{b}\right\}^\top \boldsymbol{A}_{\eta_n}\boldsymbol{H}\boldsymbol{A}_{\eta_n}\left\{n^{-1/2}\boldsymbol{\xi}_n + \eta_n\boldsymbol{b}\right\}\right] + o(a_n).$$

Because $E(\boldsymbol{\xi}_n) = \boldsymbol{0}$, the cross term vanishes. Moreover, $E(\boldsymbol{\xi}_n\boldsymbol{\xi}_n^\top) \to \boldsymbol{\Sigma}$. Therefore,

$$E\left[R(\widehat{\boldsymbol{\theta}}_{\eta_n}) - R(\boldsymbol{\theta}^\star)\right] = \frac{1}{2n}\operatorname{tr}\left(\boldsymbol{H}\boldsymbol{A}_{\eta_n}\boldsymbol{\Sigma}\boldsymbol{A}_{\eta_n}\right) + \frac{\eta_n^2}{2}\boldsymbol{b}^\top \boldsymbol{A}_{\eta_n}\boldsymbol{H}\boldsymbol{A}_{\eta_n}\boldsymbol{b} + o(a_n),$$

which proves (11).

**Step 4: Small-$\eta$ expansion.** Using the matrix expansion

$$(\boldsymbol{H} + \eta\boldsymbol{G})^{-1} = \boldsymbol{H}^{-1} - \eta\boldsymbol{H}^{-1}\boldsymbol{G}\boldsymbol{H}^{-1} + O(\eta^2),$$

we obtain

$$\boldsymbol{A}_\eta = \boldsymbol{H}^{-1} - \eta\boldsymbol{H}^{-1}\boldsymbol{G}\boldsymbol{H}^{-1} + O(\eta^2).$$

Substituting this expansion into the variance term gives

$$\frac{1}{2n}\operatorname{tr}\left(\boldsymbol{H}\boldsymbol{A}_{\eta_n}\boldsymbol{\Sigma}\boldsymbol{A}_{\eta_n}\right) = \frac{1}{2n}\operatorname{tr}\left(\boldsymbol{H}^{-1}\boldsymbol{\Sigma}\right) - \frac{\eta_n}{n}\operatorname{tr}\left(\boldsymbol{H}^{-1}\boldsymbol{G}\boldsymbol{H}^{-1}\boldsymbol{\Sigma}\right) + O\left(\frac{\eta_n^2}{n}\right).$$

Similarly,

$$\frac{\eta_n^2}{2}\boldsymbol{b}^\top \boldsymbol{A}_{\eta_n}\boldsymbol{H}\boldsymbol{A}_{\eta_n}\boldsymbol{b} = \frac{\eta_n^2}{2}\boldsymbol{b}^\top \boldsymbol{H}^{-1}\boldsymbol{b} + O(\eta_n^3).$$

Combining the two displays yields (12). The expansion identifies the leading variance-reduction and teacher-induced bias terms, but its remainder is not sharp enough to establish a strict sign for the risk difference at $\eta_n = O(n^{-1})$ without additional higher-order assumptions. This completes the proof. $\qquad\square$

## B. Collapsed Single-Risk Distillation

In some applications, interest centers on a single event type (e.g. disease-specific death), while other competing risks are infrequent or uninformative. In such settings, it is common to treat non-primary risks as censoring and model the cause-specific hazard for the event of interest (Lee et al., 2018b; Wu et al., 2022). We include this variant for completeness, but do not use it as the main formulation because the main text focuses on competing-risks students.

Assume $j = 1$ denotes the primary event of interest. The "one-vs-all" formulation is

$$L = \prod_{i=1}^{n}\prod_{k=1}^{K}\left[\lambda_1(\tau_k; \boldsymbol{Z}_i)^{\delta_{ik}^{(1)}}\{1 - \lambda_1(\tau_k; \boldsymbol{Z}_i)\}^{1-\delta_{ik}^{(1)}}\right]^{Y_{ik}}. \tag{13}$$

This can be viewed as a composite conditional likelihood for the primary cause-specific hazard. It targets the cause-specific hazard without requiring independence among competing risks. In contrast, treating competing risks as censoring and interpreting the resulting model as a marginal hazard model is generally invalid when competing risks are correlated.

Given that both student and teacher focus on the same primary event of interest, knowledge can be distilled through the discrete-time hazard-based KL divergence:

$$d(\tilde{P}\|P_{\boldsymbol{r}};\tau_k, \boldsymbol{Z}_i) = \tilde{\lambda}_1(\tau_k; \boldsymbol{Z}_i)\log\left(\frac{\tilde{\lambda}_1(\tau_k; \boldsymbol{Z}_i)}{\lambda_1(\tau_k; \boldsymbol{Z}_i)}\right)$$

$$+ \{1 - \tilde{\lambda}_1(\tau_k; \boldsymbol{Z}_i)\}\log\left(\frac{1 - \tilde{\lambda}_1(\tau_k; \boldsymbol{Z}_i)}{1 - \lambda_1(\tau_k; \boldsymbol{Z}_i)}\right). \tag{14}$$

Integrating the corresponding accumulated KL divergence into the internal log-composite likelihood yields

$$\ell_\eta(\boldsymbol{r}) \propto \sum_{i=1}^{n} \sum_{k=1}^{K} Y_{ik} \left[ \frac{\delta_{ik}^{(1)} + \eta\, \tilde{\lambda}_1(\tau_k; \boldsymbol{Z}_i)}{1 + \eta} \left\{ \log \lambda_1(\tau_k; \boldsymbol{Z}_i) - \log\{1 - \lambda_1(\tau_k; \boldsymbol{Z}_i)\} \right\} \right.$$
$$\left. + \log\{1 - \lambda_1(\tau_k; \boldsymbol{Z}_i)\} \right]. \tag{15}$$

Thus, the collapsed single-risk variant has the same soft-label interpretation as the main DiSKD objective, but operates on the Bernoulli interval outcome for the primary cause rather than the full competing-risks outcome space.

## C. Evaluation Metrics

We evaluate model performance using both predictive accuracy and explanation-based interpretability. Specifically, we report $C^{td}$, IBS, IBLL, and the predictive deviance as primary metrics. To assess variable importance and model interpretability, we compute SHAP values and use the Jaccard index to quantify feature selection stability.

### C.1. Time-dependent Concordance Index

The time-dependent concordance index ($C^{td}$) (Antolini et al., 2005), quantifies the model's ability to rank survival probabilities in accordance with actual event times for a random pair of observations:

$$C^{td} = P\left( \hat{S}(X_i | \boldsymbol{Z}_i) < \hat{S}(X_i | \boldsymbol{Z}_j) | X_i < X_j, \Delta_i = 1 \right)$$
$$\approx \frac{\sum_{i \neq j} \mathbb{1}[X_i < X_j] \cdot \mathbb{1}\left[ \hat{S}(X_i | \boldsymbol{Z}_i) < \hat{S}(X_i | \boldsymbol{Z}_j) \right] \cdot \delta_i}{\sum_{i \neq j} \mathbb{1}[X_i < X_j] \cdot \delta_i}, \tag{16}$$

where $X_i = \min(D_i, C_i)$ is the observed time, $\hat{S}$ denotes predicted survival probabilities, and $\delta_i = \mathbb{1}(\Delta_i = 1)$. Note that the $C^{td}$ reduces to the standard concordance index ($C$-index) (Harrell et al., 1982) when the proportional hazards assumption holds.

In the competing risks scenario, we follow the extended definition of the $C^{td}$ for risk $k$ as described by Lee et al. (2018a):

$$C_k^{td} = P\left( \hat{F}_k(X_i | \boldsymbol{Z}_i) > \hat{F}_k(X_i | \boldsymbol{Z}_j) | X_i < X_j, \Delta_i = k \right)$$
$$\approx \frac{\sum_{i \neq j} A_{k,i,j} \cdot \mathbb{1}\left[ \hat{F}_k(X_i | \boldsymbol{Z}_i) > \hat{F}_k(X_i | \boldsymbol{Z}_j) \right]}{\sum_{i \neq j} A_{k,i,j}}, \tag{17}$$

where $\hat{F}_k$ denotes predicted cumulative incidence function for risk $k$ and

$$A_{k,i,j} = \mathbb{1}(\Delta_i = k, X_i < X_j) \tag{18}$$

capturing pairs where one individual experiences event $k$ while the other individual remains at risk without having experienced the event. A higher $C^{td}$ indicates better discrimination.

### C.2. Integrated Brier Score (IBS)

The Integrated Brier Score (IBS) evaluates the global calibration and accuracy of time-dependent survival predictions under right censoring. Following the IPCW (inverse probability of censoring weighted) formulation from Gerds & Schumacher (2006); Graf et al. (1999); Kvamme et al. (2019), the Brier score at time $t$ is defined as

$$\text{BS}(t) = \frac{1}{n} \sum_{i=1}^{n} W_i(t) \left( \hat{S}(t \mid \boldsymbol{Z}_i) - \mathbb{1}(X_i > t) \right)^2, \tag{19}$$

where $\hat{S}(t \mid \boldsymbol{Z}_i)$ denotes the predicted survival probability and $W_i(t)$ represents the IPCW weight constructed from the Kaplan–Meier estimator of the censoring distribution. This weighting corrects for informative loss of individuals due to censoring and yields an unbiased estimate of prediction error.

The IBS aggregates the Brier score over the full time interval:

$$\text{IBS} = \frac{1}{t_{\max} - t_{\min}} \int_{t_{\min}}^{t_{\max}} \text{BS}(t) \, dt, \tag{20}$$

which is approximated via numerical integration. A lower IBS indicates better calibration and discrimination of survival predictions across time.

### C.3. Integrated Binomial Log-Likelihood (IBLL)

Another metric evaluating both calibration and discriminative performance is the Integrated Binomial Log-Likelihood (IBLL), based on the IPCW-weighted binomial log-likelihood introduced by Graf et al. (1999); Kvamme et al. (2019). Let $\hat{S}(t \mid \boldsymbol{Z}_i)$ denote the predicted survival probability at time $t$ and $\hat{G}(t)$ the Kaplan–Meier estimator of the censoring distribution. The IPCW binomial log-likelihood at time $t$ is

$$\text{BLL}(t) = \frac{1}{n} \sum_{i=1}^{n} \left[ \frac{\log\left(1 - \hat{S}(t \mid \boldsymbol{Z}_i)\right) \, \mathbb{1}(X_i \leq t, \Delta_i = 1)}{\hat{G}(X_i)} + \frac{\log\left(\hat{S}(t \mid \boldsymbol{Z}_i)\right) \, \mathbb{1}(X_i > t)}{\hat{G}(t)} \right]. \tag{21}$$

Following the common implementation convention in survival prediction software, we report the negative integrated binomial log-likelihood:

$$\text{IBLL} = -\frac{1}{t_{\max} - t_{\min}} \int_{t_{\min}}^{t_{\max}} \text{BLL}(t) \, dt. \tag{22}$$

Compared with the squared-error penalty used in the Brier score, the binomial log-likelihood uses a logarithmic scoring rule and therefore penalizes confident but incorrect survival probability predictions more strongly. Since we report the negative integrated value, lower reported IBLL values indicate better calibration and likelihood fit of the predicted discrete-time survival distributions.

### C.4. Predictive Deviance

Another metric is the predictive deviance (Burnham & Anderson, 2002; Tutz et al., 2016), defined as the negative log-likelihood of the fitted model on the test data, reflecting the discrepancy between the learned and the true data-generating processes.

### C.5. SHAP

SHAP (SHapley Additive exPlanations) is a widely adopted technique in Explainable Artificial Intelligence (XAI) that attributes the prediction of a model to its input features based on cooperative game theory. It approximates Shapley values—a principled way of distributing a model's output among its input features based on their marginal contributions (Lundberg & Lee, 2017; Vo et al., 2024). Formally, for a predictive model $v$ that outputs $v(\boldsymbol{Z}_i)$ for an input $\boldsymbol{Z}_i = (Z_{i1}, Z_{i2}, \ldots, Z_{ip})$, the Shapley value $\phi_j$ corresponding to the $j$th feature is defined as

$$\phi_j = \sum_{S \subseteq P \setminus \{j\}} \frac{|S|!(p - |S| - 1)!}{p!} \left[ v(S \cup \{j\}) - v(S) \right], \tag{23}$$

where $P = \{1, 2, \ldots, p\}$ denotes the index set of features, and $S$ is a subset of $P$ that excludes $j$. The expression quantifies the expected marginal improvement in the prediction by including feature $j$ over all possible feature coalitions. These values offer consistent and model-agnostic explanations, making them particularly suitable for interpreting complex models like deep neural networks.

In practice, we compute the mean absolute SHAP value for each feature across all individuals and rank the features by their average importance. The top $d$ features in this ranking are designated as the most important variables (Marcílio & Eler, 2020). This allows us to capture not just which features contribute most on average, but also to compare how different models prioritize variables under varying data configurations.

## C.6. Jaccard Index

Model training and data splitting introduce randomness that may affect the stability of variable selection. To address this, we quantify the agreement in selected important features across repeated runs using the Jaccard index. Given two feature sets $A$ and $B$ (e.g. top-20 SHAP-ranked variables from two replicates), the Jaccard index is defined as

$$\text{Jaccard}(A, B) = \frac{|A \cap B|}{|A \cup B|}, \tag{24}$$

which ranges from 0 (no overlap) to 1 (complete overlap). A higher Jaccard index implies that the model consistently identifies similar sets of influential variables across different random seeds or data partitions, thus enhancing the credibility of feature-based explanations.

# D. Additional Experimental Settings

## D.1. Additional Implementation Details

**Implementation.** We implemented DiSKD in Python 3.9 with PyTorch 1.12.1. Each experiment follows a two-stage pipeline: we first train a teacher model on the external dataset, and then train a student model on the internal dataset with a KL-regularized objective that distills the teacher predictions. To ensure comparability, we use the same training, validation, and tuning protocol across all experiments. Unless otherwise noted, we use a Transformer-based discrete-time survival network as the default architecture. All training runs last for at most 128 epochs with early stopping: training stops if the validation loss does not improve for 5 consecutive epochs, where an epoch denotes one full pass over the training data.

**Tuning parameters.** We tune architectural and optimization hyperparameters by 5-fold cross-validation using validation predictive deviance. The search space includes the number of layers $\{1, 2, 3, 4\}$, hidden units per layer $\{32, 64, 128\}$, and dropout rate $[0.1, 0.2]$, together with batch size $\{32, 64, 128\}$ and learning rate $[5 \times 10^{-4}, 10^{-3}]$. Batch normalization is enabled in all runs. We also tune the distillation temperature $T$ over $(0, 5)$ and the distillation weight $\eta$ over $[0, 10]$.

We use `Optuna` for automated hyperparameter optimization (Akiba et al., 2019), with the Tree-structured Parzen Estimator (TPE) sampler (Bergstra et al., 2011; 2013; Watanabe, 2023). TPE maintains separate density models for high-performing and low-performing configurations from previous trials and proposes new candidates that maximize the expected improvement of the validation objective. Compared with manual grid search, this Bayesian optimization procedure avoids rigid discretization and can treat $\eta$ as a continuous parameter, which is particularly useful in small-sample settings where coarse grids may yield unstable or sub-optimal choices. We run 20 trials per optimization run and select the configuration with the lowest validation predictive deviance, balancing search thoroughness with computational cost.

Because the optimal $\eta$ depends on the relative informativeness of the teacher predictions versus the internal data, there is no single fixed default that is universally appropriate. `Optuna` addresses this by letting us specify a wide initial search range for $\eta$ rather than committing to a hand-crafted grid or a single recommended value. This removes the burden of choosing grid resolution and coverage: practitioners can initialize the tuning pipeline with a broad range, and the Bayesian optimizer will adaptively concentrate its search and converge to an appropriate weighting for the dataset at hand.

For the teacher, we split the external dataset into training and validation subsets, perform TPE-based 5-fold cross-validation to select hyperparameters, and then refit the final teacher under the selected configuration. For the student, we considered two strategies: (i) jointly tuning network hyperparameters together with $(T, \eta)$ in a single cross-validation loop, and (ii) first tuning the network hyperparameters and then selecting $(T, \eta)$ in an additional cross-validation step. Both strategies produced comparable performance in preliminary experiments; we adopt the sequential strategy due to its lower computational cost.

## D.2. Simulation Settings

**Data-generating process.** We consider a competing-risks setting with two correlated causes. We generate 12 independent covariates $\boldsymbol{X} = (X_1, \ldots, X_{12})^\top$ with $X_p \sim \mathcal{N}(0, 1)$ and define three linear summaries $Z_1 = \sum_{p=1}^{4} X_p$, $Z_2 = \sum_{p=5}^{8} X_p$, and $Z_3 = \sum_{p=9}^{12} X_p$. Latent failure times are sampled from exponential distributions whose rate parameters depend nonlinearly on $(Z_1, Z_2, Z_3)$:

$$D_1 \sim Exp\left\{(\beta_1 Z_1)^2 + (\beta_3 Z_3)^2\right\},$$
$$D_2 \sim Exp\left\{(\beta_2 Z_2)^2 + (\beta_3 Z_3)^2\right\}.$$

Here, $X_1$ to $X_4$ are associated only with risk 1, $X_5$ to $X_8$ only with risk 2, while $X_9$ to $X_{12}$ influence both. Independent censoring times were sampled from a uniform distribution $C \sim \mathcal{U}(0, 500)$. The failure time and event indicators were constructed as

$$D = \min\{D_1, D_2\},$$
$$\Delta = \begin{cases} \arg\min\{D_1, D_2\} & \text{if } D < C, \\ 0 & \text{otherwise.} \end{cases}$$

Observed times were discretized into 20 intervals using Kaplan–Meier quantiles. With parameters $\beta_1 = 2$, $\beta_2 = 2$, and $\beta_3 = 8$, this setup produced event rates of approximately 29.76% for risk 1 and 29.36% for risk 2.

In this design, the two competing risks are highly correlated. This arises because both cause-specific failure times share the same dominant nonlinear term $(\beta_3 Z_3)^2$ in their rate functions. Although $Z_1$ and $Z_2$ uniquely affect risks 1 and 2, respectively, the magnitude of $(\beta_3 Z_3)^2$ is much larger when $\beta_3$ is set to a large value, and therefore this shared component explains the majority of the variation in the event times for both causes. As a result, individuals with large values of $Z_3$ simultaneously experience similar shifts in the risk levels of both causes. Consequently, the two cause-specific hazards tend to move in parallel, leading to strong dependence between the two competing risks.

We generate a teacher-training dataset ($n = 10,000$), an internal student-training dataset ($n = 500$), and an independent test dataset ($n = 5,000$) from the same distribution. All experiments are repeated over 20 random seeds.

**Methods compared.** We compare an internal-only baseline with two DiSKD variants that differ in the form of teacher information: (1) Internal: a competing-risks student trained on the internal dataset using all covariates, without distillation; (2) DiSKD-C: distillation from a competing-risks teacher that provides cause-specific discrete-time hazard predictions for each risk; (3) DiSKD-O: distillation from an overall-event teacher that provides only the aggregated hazard for the composite event. To vary teacher quality, we restrict the covariates available to the teacher while keeping the student covariates fixed. We consider three teacher configurations: (a) Good-quality teacher: used the full covariate set; (b) Fair-quality teacher: used a subset of covariates $\{X_1, X_2, X_3, X_5, X_6, X_7, X_9, X_{10}, X_{11}\}$; (c) Poor-quality teacher: used a smaller subset $\{X_1, X_2, X_5, X_6, X_9, X_{10}\}$.

### D.3. Benchmark Survival Data Settings

D.3.1. SUPPORT

The Study to Understand Prognoses and Preferences for Outcomes and Risks of Treatment (SUPPORT) is a large-scale study examining survival outcomes among seriously ill hospitalized patients (Knaus et al., 1995). Approximately 68.1% of participants died during the study, with a median survival time of 58 days. The original dataset includes 9,105 patients and 14 covariates. Due to varying admission criteria and screening procedures, individuals entered the study at different time points.

The covariates comprise demographic characteristics (age, sex, race) and clinical measurements recorded on the third day of the study (or the first day if third-day data were unavailable). These include the number of comorbidities, and the presence of diabetes, dementia, and cancer, along with physiological measures such as mean arterial blood pressure, heart rate, respiratory rate, temperature, white blood cell count, serum sodium, and serum creatinine. Patients who died before the third study day were excluded, ensuring nearly complete data for the remaining participants. After excluding records with missing values, 8,873 patients were retained for analysis.

For our experiments, we used the pre-processed version of the dataset provided by (Katzman et al., 2018). We further assumed a maximum follow-up period of 180 days, treating patients surviving beyond this period as censored, which resulted in a censoring rate of 53%. The follow-up period was divided into 20 evenly spaced intervals, and each patient's event time was assigned to a corresponding interval. Alternatively, interval boundaries can be defined using quantiles of the Kaplan-Meier estimator. This preprocessing step prepares the data for modeling using discrete-time survival methods, which are particularly suitable for short follow-up periods and densely spaced time points.

### D.3.2. METABRIC

The Molecular Taxonomy of Breast Cancer International Consortium (METABRIC) dataset is a widely used breast cancer survival benchmark that combines molecular profiles with clinical features to study disease subtypes and prognosis (Curtis et al., 2012). It contains gene expression measurements and clinical variables for 1,980 patients, among whom 57.72% have an observed death due to breast cancer, with a median survival time of 116 months. Following standard preprocessing used in prior work, we construct covariates in line with the Immunohistochemical 4 plus Clinical (IHC4+C) setting by combining four gene indicators (MKI67, EGFR, PGR, and ERBB2) with clinical features, including treatment indicators (hormone therapy, radiotherapy, and chemotherapy), ER status, and age at diagnosis. In our experiments, we discretize follow-up time into 20 intervals.

### D.3.3. GBSG

We use the combined Rotterdam tumor bank and German Breast Cancer Study Group (Rotterdam & GBSG) benchmark, which contains $n = 2,232$ patients with no missing values (Foekens et al., 2000; Schumacher et al., 1994). In total, 965 (43.23%) individuals are right-censored. Each patient has 7 covariates, including hormone therapy, age, menopausal status, tumor grade, number of positive nodes, progesterone receptor, and estrogen receptor. In our experiments, we discretize follow-up time into 20 intervals.

### D.4. SRTR Model Updating Settings

This study used data from the Scientific Registry of Transplant Recipients (SRTR). The SRTR data system includes data on all donor, wait-listed candidates, and transplant recipients in the US, submitted by the members of the Organ Procurement and Transplantation Network (OPTN). The Health Resources and Services Administration (HRSA), U.S. Department of Health and Human Services provides oversight to the activities of the OPTN and SRTR contractors. We analyzed one-year risks of death and graft failure among adult kidney transplant recipients.

**Cohort construction and eras.**   To separate the effects of overlapping clinical and policy changes, we divided the data into two distinct eras: the COVID period (March 1, 2020–May 11, 2023; $n = 52,436$) and the post-COVID period (May 11–September 1, 2023; $n = 5,041$). The earlier period spans both the COVID-19 pandemic and the national rollout of the KAS250 allocation policy, during which transplant volumes, donor characteristics, and recipient outcomes underwent major structural shifts. In contrast, the post-COVID period represents a stabilized phase of the U.S. transplant system under the full implementation of KAS250, providing the most relevant window for evaluating post-policy outcomes and for developing models aligned with current practice.

**Follow-up and evaluation window.**   The SRTR registry is released through periodic snapshots that add new transplants and may revise historical records. Because our evaluation targets one-year outcomes, we restrict the internal cohort to the post-COVID cohort from May 11, 2023 to September 1, 2023 for which complete one-year follow-up was available at the time of analysis.

**Teacher–student updating protocol.**   We treat the post-COVID cohort as the internal target population and the COVID+KAS250 cohort as a large but distributionally shifted historical source. Crucially, in many registry workflows the transferable artifact is a previously developed model (or its predictions), rather than sharable individual-level data. To reflect this deployment reality, we train a teacher model on the COVID+KAS250 cohort and use its discrete-time cause-specific hazard predictions as the supervision signal for DiSKD when fitting a student on the post-COVID cohort. This protocol operationalizes model updating under real temporal and policy shifts while avoiding any requirement to pool individual-level data across eras.

### D.5. SRTR Donor Risk Prediction Settings

**Teacher–student setup.**   We treat as the teacher a DiSKD-updated Transformer-based discrete-time competing-risks model trained to predict one-year death and graft failure in the post-COVID cohort (Section 5.5). The teacher leverages the full donor, recipient, and transplant-center covariate set and is updated using guidance learned from the COVID-era cohort, producing cause-specific hazard sequences for both endpoints. We then distill these predictions into a donor-only student, implemented as a lightweight two-layer Transformer with 32 hidden dimensions. This design reflects a deployment

constraint: the student is restricted to donor information available at organ offer, while the teacher captures richer clinical context.

**Donor-only input configurations.**   To connect with realistic implementation targets, we evaluate two donor feature sets. In a maximal-donor setting, the student uses all donor-related variables available in SRTR, representing the strongest donor-only predictor when a comprehensive donor record can be accessed at scoring time. In a KDRI-compatible setting, the student is restricted to the eight donor factors used by the Kidney Donor Risk Index (KDRI): donor age, height, weight, hypertension, diabetes, cause of death (CVA vs. other), terminal creatinine, and DCD status. This restricted setting is operationally important because KDRI is the standard donor-only interface used in U.S. kidney allocation; improvements achieved without changing required inputs are more readily translatable than proposals that require additional variables. Accordingly, we include KDRI as a donor-only benchmark.

**Baselines and evaluation.**   To isolate the value of distillation, we compare the distilled donor-only student against (i) an internal donor-only model with the same architecture trained directly on the post-COVID cohort without teacher guidance and (ii) KDRI. All models are trained and evaluated using 5-fold cross-validation repeated 10 times.

## E. Additional Experimental Analysis

### E.1. Simulation

#### E.1.1. EFFECT OF DISTILLATION WEIGHT $\eta$

The distillation weight $\eta$ controls the strength of KL guidance relative to the student likelihood and is therefore a key practical tuning parameter in DiSKD. With an informative teacher, increasing $\eta$ can amplify useful guidance and improve generalization in small internal cohorts. With a weak or misspecified teacher, excessively large $\eta$ can over-emphasize unreliable guidance and lead to degraded performance. We therefore sweep $\eta$ to assess sensitivity and to understand how the preferred distillation strength varies with teacher quality (as induced by covariate restriction).

Figure 5 summarizes performance as a function of $\eta$. When the teacher is strong, DiSKD-C achieves the best results across a wide range of $\eta$ in both $C^{td}$ and predictive deviance, indicating stable gains from distilling cause-specific hazards. As teacher quality degrades, the advantage of DiSKD-C over DiSKD-O shrinks and the curves converge, suggesting that when teacher information is limited, distilling the aggregated hazard captures most of the transferable signal and additional cause-specific structure yields smaller marginal benefit.

#### E.1.2. DISTILLATION UNDER COVARIATE SHIFT

To assess robustness when the teacher is learned from a population that is distributionally shifted relative to the student, we conduct a covariate-shift sensitivity analysis. We generate two teacher-training datasets representing moderate and severe shift, while keeping the student-training and test datasets identical to Appendix D.2.

We retain the same covariate partition as Appendix D.2, with $\boldsymbol{X} = (X_1, \ldots, X_{12})^\top$ grouped into three blocks $\boldsymbol{X}_{1:4}$, $\boldsymbol{X}_{5:8}$, and $\boldsymbol{X}_{9:12}$. For each block $g \in \{1, 2, 3\}$, we draw an independent binary mask $\boldsymbol{m}_g \in \{0, 1\}^4$ with i.i.d. entries $m_{g\ell} \sim Bernoulli(p)$ and sample

$$\boldsymbol{X}_g \sim \mathcal{N}(\boldsymbol{m}_g \cdot s, \ \boldsymbol{I}_4),$$

where $s$ controls the shift magnitude and $p$ controls the expected fraction of shifted coordinates within each block.

Using the same linear summaries $Z_1 = \sum_{p=1}^{4} X_p$ and $Z_2 = \sum_{p=5}^{8} X_p$, we generate cause-specific latent failure times from exponential distributions, but with a different nonlinear component than in Appendix D.2. Specifically, instead of the squared linear combination $(\beta_3 Z_3)^2$ (with $Z_3 = \sum_{p=9}^{12} X_p$), we use a separable sum-of-squares term:

$$D_1 \sim Exp\left\{ (\beta_1 Z_1)^2 + \sum_{p=9}^{12} (\beta_3 X_p)^2 \right\},$$

$$D_2 \sim Exp\left\{ (\beta_2 Z_2)^2 + \sum_{p=9}^{12} (\beta_3 X_p)^2 \right\}.$$

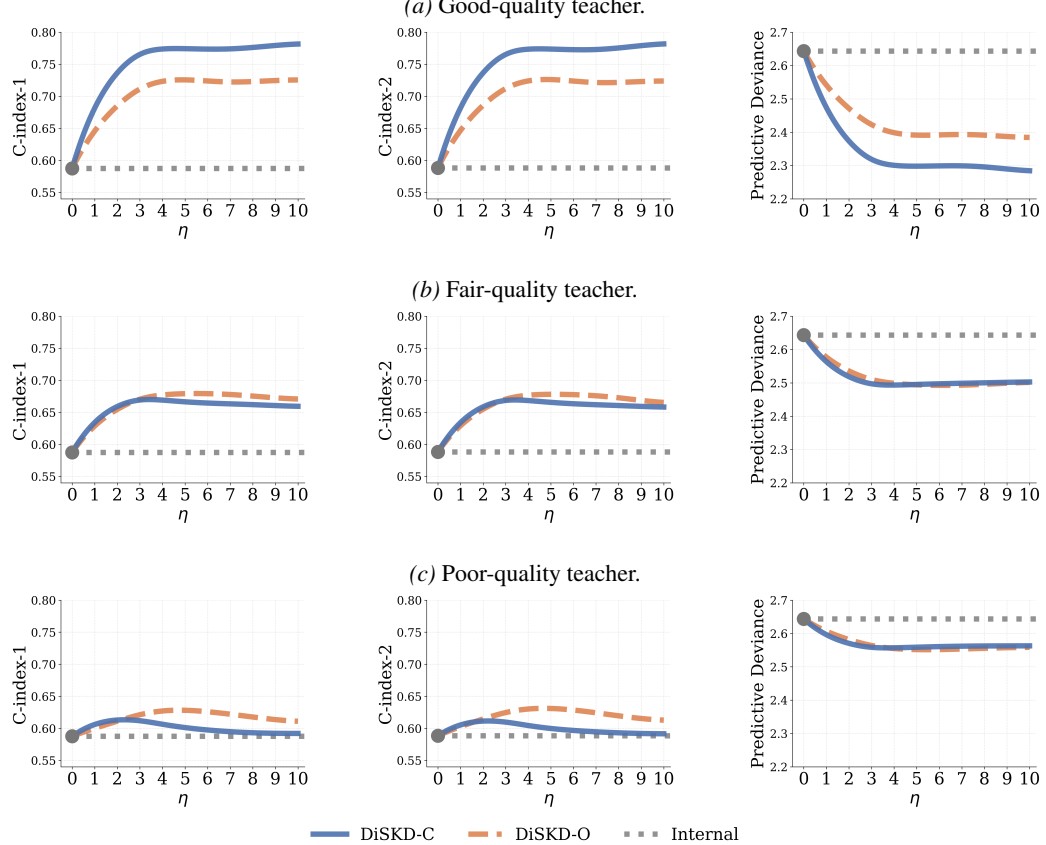

*Figure 5.* Test set $C^{td}$ and predictive deviance across varying values of the distillation weight $\eta$ under different teacher quality settings (a–c) for simulated competing risks data. We compare three modeling strategies: (1) Internal: a Transformer-based discrete-time competing risks model trained on the internal dataset using all covariates; (2) DiSKD-C: our KL-based knowledge distillation using teacher discrete-time cause-specific hazard predictions from a Transformer-based competing risks model; (3) DiSKD-O: a variation of our KL-based knowledge distillation using teacher discrete-time overall hazard predictions from a Transformer-based overall event (any risk) model. Each panel (a–c) corresponds to a different teacher quality level, reflecting varying degrees of covariate availability. Results were averaged over 20 random replicates of simulated competing risks data. Higher $C^{td}$ and lower predictive deviance indicate better predictive performance.

We then sample $C \sim \mathcal{U}(0, 500)$ and construct $(D, \Delta)$ as in Appendix D.2, with follow-up discretized into 20 intervals using Kaplan–Meier quantiles. Unless otherwise noted, we use $(\beta_1, \beta_2, \beta_3) = (2, 2, 8)$.

We generate two teacher-training datasets ($n = 10{,}000$ each): (i) moderate shift with $(s, p) = (-0.3, 0.3)$ and (ii) severe shift with $(s, p) = (1.0, 0.6)$. These regimes induce different censoring and incidence levels: the moderate-shift teacher population has event rates 23.26% (risk 1) and 22.73% (risk 2), while the severe-shift population has event rates 16.19% (risk 1) and 16.36% (risk 2). In both cases, the teacher is trained on the shifted population and the student is trained on the original internal cohort using DiSKD, isolating the effect of teacher–student covariate shift on knowledge transfer.

Figure 6 shows that DiSKD remains effective under both moderate and severe shift, improving $C^{td}$ for both risks and reducing predictive deviance relative to the internal-only student. Notably, even when a shifted teacher is poorly calibrated on the student test distribution, distillation does not induce systematic degradation, consistent with robustness to covariate shift.

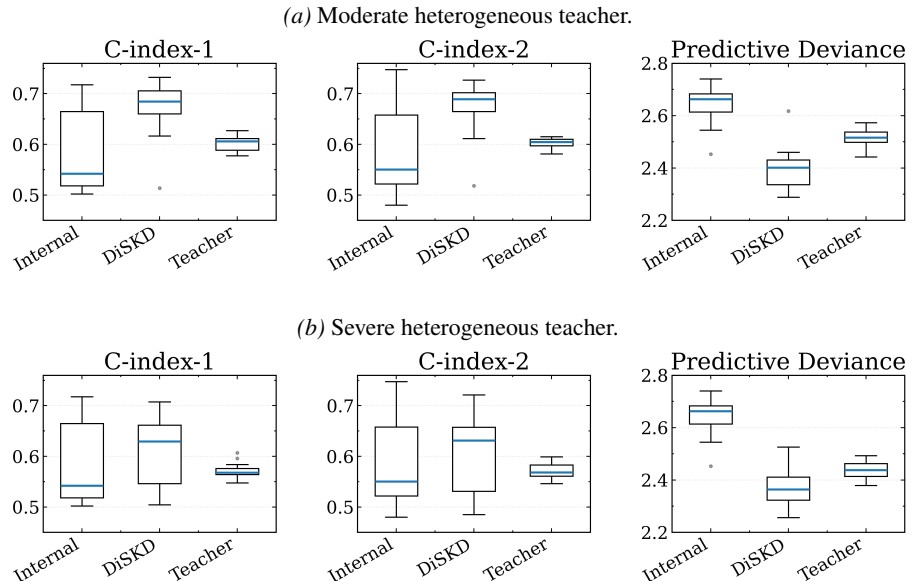

*Figure 6.* Robust distillation under heterogeneous teacher-training datasets. Boxplots summarize test-set $C^{td}$ for each risk and predictive deviance when the teacher is trained on a distributionally shifted external cohort with (a) moderate and (b) severe heterogeneity. We compare Internal baseline, DiSKD, and the shifted Teacher evaluated on the same internal test set, highlighting that DiSKD improves over Internal and avoids negative transfer even when the teacher is misaligned.

### E.1.3. MULTI-TEACHER DISTILLATION

We further evaluate whether DiSKD can combine information from multiple teachers with different levels of compatibility. This setting is motivated by applications where several external models are available, but their relevance to the internal target cohort may differ due to covariate availability, model quality, or population heterogeneity. Suppose $M$ teachers provide predictions that can be mapped to the student's interval-level outcome space. The single-teacher KL penalty can be extended to a weighted sum of teacher-specific KL penalties:

$$\ell_{\boldsymbol{\eta}}(\boldsymbol{r}) = \ell(\boldsymbol{r}) - \sum_{m=1}^{M} \eta_m \mathrm{D}(\tilde{P}_m \| P_{\boldsymbol{r}}),$$

where $\eta_m \geq 0$ controls the strength of guidance from teacher $m$.

We conduct a controlled two-teacher simulation with one strong teacher and one weak teacher. The good teacher is trained with more informative covariates and achieves substantially better predictive performance, whereas the bad teacher is trained under degraded covariate information and is only weakly informative. We compare the internal-only student, students distilled from each teacher separately, the jointly distilled student, and the two teacher models evaluated on the same test set. For the joint model, $\eta_{\mathrm{good}}$ and $\eta_{\mathrm{bad}}$ are selected by internal validation.

Table 2 summarizes the results. The selected weights behave sensibly: the joint model assigns a larger weight to the good teacher than to the bad teacher. Compared with distillation from the bad teacher alone, joint distillation substantially improves both discrimination and predictive deviance. Compared with distillation from the good teacher alone, joint distillation gives slightly higher $C^{td}$ for both risks, although its predictive deviance is somewhat worse. Overall, the jointly distilled student remains much closer to the good-teacher solution than to the bad-teacher solution, suggesting that the validation-selected KL weights can emphasize the more compatible teacher while limiting the influence of weaker guidance.

*Table 2.* Controlled simulation results for the multiple-teacher setting. We report the internal baseline, students distilled from the good teacher, the bad teacher, or both teachers jointly, and the two teacher models themselves. For the joint-distillation model, $\eta_{\mathrm{good}}$ and $\eta_{\mathrm{bad}}$ denote the selected weights for the two teachers. Each entry is reported as mean (standard deviation).

| Method | $\eta_{\mathrm{good}}$ | $\eta_{\mathrm{bad}}$ | $C^{td}$ (risk 1) | $C^{td}$ (risk 2) | Predictive deviance |
|---|---|---|---|---|---|
| Internal | – | – | 0.5628 (0.0920) | 0.5603 (0.0892) | 2.6399 (0.0874) |
| *Students* | | | | | |
| Student (good teacher) | 6.3500 (2.4554) | – | 0.6958 (0.1294) | 0.6979 (0.1270) | 2.4247 (0.1903) |
| Student (bad teacher) | – | 2.5000 (1.6059) | 0.5789 (0.0452) | 0.6017 (0.0597) | 2.8910 (0.0548) |
| Student (both teachers) | 6.7000 (2.2965) | 2.2000 (1.5424) | 0.7039 (0.0704) | 0.7065 (0.0719) | 2.4895 (0.1033) |
| *Teachers* | | | | | |
| Teacher (good) | – | – | 0.7936 (0.0073) | 0.7924 (0.0069) | 2.2167 (0.0302) |
| Teacher (bad) | – | – | 0.5814 (0.0115) | 0.5960 (0.0062) | 2.9363 (0.0318) |

## E.2. Benchmark Survival Data

### E.2.1. TRAINING SAMPLE SIZE EFFECT ON SUPPORT

We examine how internal sample size affects predictive accuracy and stability on SUPPORT. We hold out a fixed test set (20%, $n = 1,765$) and vary the training size from 50 to 7,000 by subsampling the remaining data. For each training size, we fit a discrete-time Logistic Hazard model and evaluate on the same test set. Figure 7 reports the mean and standard deviation over 20 replicates. Performance improves and variability decreases monotonically with larger training sets, highlighting the small-sample fragility of discrete-time survival learning and motivating knowledge transfer when internal cohorts are limited.

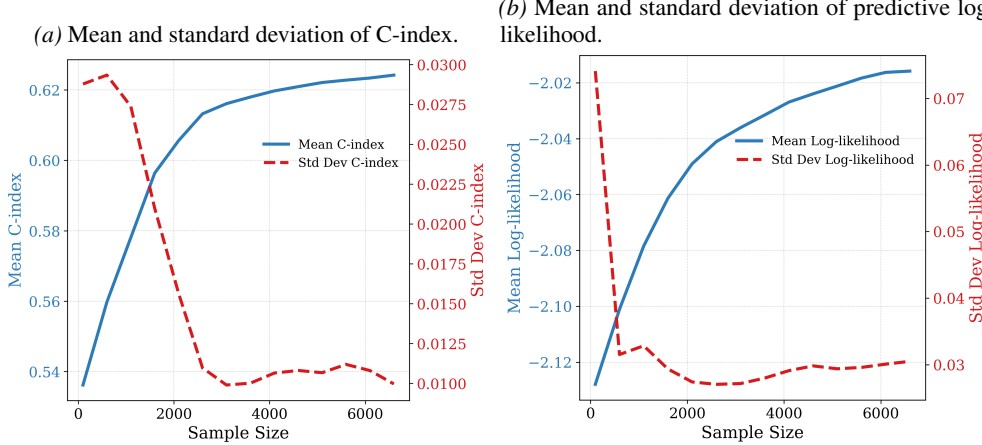

*(a)* Mean and standard deviation of C-index.  *(b)* Mean and standard deviation of predictive log-likelihood.

*Figure 7.* Test set performance of the discrete-time deep learning survival model across varying internal training sample sizes, without distillation. All experiments were conducted on the fixed SUPPORT dataset, with the test set held out (20%, $n = 1765$) and the training sample size varied from 50 to 7000 by subsampling the remaining data. Each point represents the mean or standard deviation over 20 random splits. Panel (a) reports the mean $C^{td}$ (left $y$-axis, solid blue) and its standard deviation (right $y$-axis, dashed red) as training size increases. Panel (b) shows the mean predictive log-likelihood (left $y$-axis, solid blue) and its standard deviation (right $y$-axis, dashed red). Larger training sample sizes generally yield improved average performance and reduced variability, demonstrating the benefit of more internal information for model stability and generalization.

### E.2.2. KNOWLEDGE DISTILLATION FROM LARGE TO SMALL COHORTS

We evaluate distillation from a large teacher cohort to a small student cohort on SUPPORT. We randomly split the data into a held-out test set (20%, $n = 1,765$), an internal (student) training set (4%, $n = 354$), and an external (teacher) training set (76%, $n = 6,712$). We report results for five discrete-time architectures: Nnet-Survival with logit and complementary log–log links, Logistic Hazard, Time-Embedded Logistic Hazard, and a Transformer model. Within each architecture, the teacher and student use the same network structure to isolate the effect of distillation from architectural differences.

To vary teacher informativeness, we construct four teacher-covariate regimes by restricting the covariates available to the teacher while keeping the student covariates fixed: (a) Excellent: all 14 covariates; (b) Good: three covariates removed; (c) Fair: seven covariates retained; (d) Poor: four covariates retained. We compare three training strategies: (i) Internal, a student trained only on the internal cohort; (ii) DiSKD, a student trained on the internal cohort with KL distillation from the teacher predictions (with $\eta$ tuned by 5-fold cross-validation on the internal training set); and (iii) Teacher, a model trained only on the external cohort (reported as a reference for how informative the teacher is on the internal test distribution).

Table 3 summarizes test-set $C^{td}$, predictive deviance, IBS, and IBLL (mean and standard deviation over 20 random splits). Across architectures, DiSKD improves over the internal-only baseline in most settings, with the largest gains under excellent/good teacher covariates and progressively smaller gains as the teacher covariate set is degraded. Notably, when the teacher is weak (fair/poor), distillation remains stable: performance typically tracks the internal baseline rather than collapsing toward the teacher-alone model, suggesting limited negative transfer from low-informativeness teacher guidance. We also observe architecture-dependent sensitivity: higher-capacity students (Transformer, Time-Embedded) tend to extract more benefit when the teacher signal is strong, while simpler students show smaller and less consistent gains under degraded teacher covariates.

To assess whether the gains in SUPPORT carry over to other disease domains, we further evaluate DiSKD on two widely used cancer survival benchmarks, METABRIC and Rotterdam & GBSG. For each dataset, we follow the same large-to-small transfer protocol as in SUPPORT, randomly splitting the data into a held-out test set (20%), an internal student cohort (4%), and an external teacher cohort (76%). Both teacher and student are implemented as Transformer-based discrete-time survival networks to keep the modeling pipeline fixed across datasets.

Table 4 reports test-set $C^{td}$, predictive deviance, IBS, and IBLL. Across both METABRIC and GBSG, DiSKD improves over the internal-only baseline on all metrics, indicating that teacher-guided KL regularization yields consistent gains beyond a single cohort. The improvements are larger on METABRIC than on GBSG, consistent with METABRIC providing a stronger and more stable teacher signal in this split regime, while the smaller GBSG cohort yields more modest but still consistent transfer gains.

Overall, these results suggest that DiSKD is not tied to a particular dataset or event profile, but provides a broadly applicable mechanism for transferring predictive structure from a teacher survival model to a student trained on a limited internal cohort.

### E.3. SRTR Model Updating

#### E.3.1. COUPLED TEMPORAL AND GEOSPATIAL SHIFT UNDER POLICY AND PANDEMIC FORCES

During the COVID period, transplant activity was shaped by two concurrent structural forces. First, the nationwide implementation of KAS250 in March 2021 expanded organ sharing within a 250-mile radius of the donor hospital, broadening geographic access but substantially modifying the donor–recipient mix and center-level practice patterns. Second, the COVID-19 pandemic disrupted all stages of transplant care, producing fluctuations in transplant volume, elevated mortality among immunosuppressed recipients, and changes in donor availability and screening. The overlap of these two influences resulted in complex population drift and outcome shift, creating a realistic setting for evaluating methods that transfer predictive information across evolving clinical environments.

As shown in Figure 3a, the pre-COVID period exhibits relatively stable one-year mortality and graft failure rates. During COVID (2020–2023), mortality increases and becomes more volatile, with a marked peak during the Alpha surge in early 2021, whereas graft failure remains comparatively stable, suggesting that patient survival was more sensitive to pandemic-related disruptions than graft viability. Following this period, outcomes transition into a new post-pandemic regime, shaped by lingering pandemic effects and recent policy changes. Overall, the trajectory underscores a non-stationary data-generating process and cautions against naively applying models trained on pooled historical data to the most current

*Table 3.* Prediction performance across different teacher quality settings (a)–(d), evaluated under five network architectures of discrete-time deep learning survival models. The SUPPORT dataset was randomly split into three parts: a test set (20%, $n = 1765$), an internal dataset (4%, $n = 354$), and an external teacher dataset (76%, $n = 6712$). Both student and teacher models used the same network architecture within each row for fair comparison. We compare three modeling strategies: (1) Internal: a discrete-time survival model trained on the internal dataset using all covariates; (2) DiSKD: our KL-based distillation approach using discrete-time hazard predictions from a teacher model with corresponding covariate-quality levels (a)–(d), tuning parameter $\eta$ selected using 5-fold cross-validation on the internal training set; (3) Teacher: a discrete-time survival model trained solely on the external dataset using all covariates. Each entry reports the mean and standard deviation (in parentheses) of the time-dependent concordance $C^{td}$, predictive deviance, integrated Brier score (IBS), and integrated binomial log-likelihood (IBLL) evaluated on the held-out test set over 20 random splits. Higher $C^{td}$ and lower values of the other three metrics indicate better predictive performance (darker colors). Overall, DiSKD improves prediction over the internal-only model across most settings, especially when teacher quality is high.

| Structures | Method | External model | | $C^{td} \uparrow$ | Pred Dev $\downarrow$ | IBS $\downarrow$ | IBLL $\downarrow$ |
| | | Setting | Quality | | | | |
|---|---|---|---|---|---|---|---|
| Nnet-Survival (Logit) | Internal | - | - | 0.544 (0.012) | 2.223 (0.053) | 0.189 (0.008) | 0.569 (0.024) |
| | DiSKD | (a) | Excellent | 0.566 (0.015) | 2.118 (0.046) | 0.168 (0.005) | 0.509 (0.013) |
| | | (b) | Good | 0.559 (0.019) | 2.125 (0.039) | 0.170 (0.005) | 0.513 (0.015) |
| | | (c) | Fair | 0.546 (0.013) | 2.132 (0.036) | 0.170 (0.004) | 0.515 (0.013) |
| | | (d) | Poor | 0.534 (0.012) | 2.126 (0.042) | 0.170 (0.003) | 0.514 (0.008) |
| | Teacher | - | - | 0.614 (0.008) | 2.032 (0.029) | 0.159 (0.003) | 0.485 (0.008) |
| Nnet-Survival (C-log-log) | Internal | - | - | 0.544 (0.014) | 2.195 (0.039) | 0.185 (0.007) | 0.557 (0.016) |
| | DiSKD | (a) | Excellent | 0.566 (0.015) | 2.110 (0.032) | 0.168 (0.004) | 0.507 (0.009) |
| | | (b) | Good | 0.556 (0.020) | 2.114 (0.032) | 0.195 (0.114) | 0.510 (0.011) |
| | | (c) | Fair | 0.548 (0.016) | 2.118 (0.031) | 0.195 (0.116) | 0.509 (0.009) |
| | | (d) | Poor | 0.528 (0.013) | 2.128 (0.030) | 0.224 (0.156) | 0.518 (0.013) |
| | Teacher | - | - | 0.617 (0.010) | 2.029 (0.027) | 0.158 (0.003) | 0.482 (0.007) |
| Logistic Hazard | Internal | - | - | 0.546 (0.020) | 2.212 (0.085) | 0.176 (0.006) | 0.541 (0.024) |
| | DiSKD | (a) | Excellent | 0.585 (0.013) | 2.061 (0.040) | 0.165 (0.004) | 0.499 (0.01) |
| | | (b) | Good | 0.575 (0.018) | 2.088 (0.051) | 0.166 (0.004) | 0.508 (0.015) |
| | | (c) | Fair | 0.563 (0.014) | 2.069 (0.034) | 0.165 (0.003) | 0.501 (0.011) |
| | | (d) | Poor | 0.545 (0.016) | 2.077 (0.030) | 0.166 (0.003) | 0.504 (0.009) |
| | Teacher | - | - | 0.623 (0.013) | 2.024 (0.028) | 0.165 (0.004) | 0.483 (0.009) |
| Time Embedding | Internal | - | - | 0.560 (0.026) | 2.104 (0.037) | 0.168 (0.004) | 0.507 (0.011) |
| | DiSKD | (a) | Excellent | 0.616 (0.011) | 2.017 (0.027) | 0.159 (0.003) | 0.483 (0.007) |
| | | (b) | Good | 0.611 (0.025) | 2.030 (0.036) | 0.160 (0.003) | 0.485 (0.008) |
| | | (c) | Fair | 0.593 (0.014) | 2.043 (0.026) | 0.161 (0.003) | 0.487 (0.007) |
| | | (d) | Poor | 0.579 (0.019) | 2.054 (0.034) | 0.163 (0.003) | 0.492 (0.008) |
| | Teacher | - | - | 0.623 (0.011) | 2.017 (0.027) | 0.158 (0.003) | 0.479 (0.009) |
| Transformer | Internal | - | - | 0.563 (0.023) | 2.088 (0.033) | 0.167 (0.005) | 0.503 (0.011) |
| | DiSKD | (a) | Excellent | 0.621 (0.012) | 2.034 (0.032) | 0.159 (0.003) | 0.483 (0.008) |
| | | (b) | Good | 0.618 (0.013) | 2.034 (0.028) | 0.159 (0.003) | 0.483 (0.008) |
| | | (c) | Fair | 0.593 (0.015) | 2.052 (0.032) | 0.162 (0.003) | 0.490 (0.008) |
| | | (d) | Poor | 0.578 (0.020) | 2.061 (0.030) | 0.163 (0.003) | 0.493 (0.008) |
| | Teacher | - | - | 0.625 (0.012) | 2.031 (0.032) | 0.158 (0.003) | 0.479 (0.008) |

*Table 4.* Prediction performance across METABRIC and GBSG, evaluated using Transformer-based discrete-time deep learning survival model. Each dataset is randomly split into test (20%), internal (4%), and external (76%) subsets. Both student and teacher models used the same network architecture within each row for fair comparison. We compare three modeling strategies: (1) Internal: a discrete-time survival model trained on the internal dataset using all covariates; (2) DiSKD: our KL-based distillation approach using discrete-time hazard predictions from a teacher model using all covariates, tuning parameter $\eta$ selected using 5-fold cross-validation on the internal training set; (3) Teacher: a discrete-time survival model trained solely on the external dataset using all covariates. Each entry reports the mean and standard deviation (in parentheses) of the time-dependent concordance $C^{td}$, predictive deviance, integrated Brier score (IBS), and negative integrated binomial log-likelihood (IBLL) evaluated on the held-out test set over 20 random splits. Higher $C^{td}$ and lower values of the other three metrics indicate better predictive performance.

| Dataset | Method | $C^{td} \uparrow$ | Pred Dev $\downarrow$ | IBS $\downarrow$ | IBLL $\downarrow$ |
|---|---|---|---|---|---|
| METABRIC | Internal | 0.602 (0.047) | 2.117 (0.090) | 0.174 (0.017) | 0.530 (0.049) |
| | DiSKD | 0.657 (0.021) | 1.972 (0.072) | 0.158 (0.011) | 0.477 (0.030) |
| | Teacher | 0.674 (0.017) | 1.928 (0.052) | 0.148 (0.008) | 0.450 (0.019) |
| GBSG | Internal | 0.607 (0.054) | 2.338 (0.095) | 0.184 (0.013) | 0.543 (0.029) |
| | DiSKD | 0.651 (0.030) | 2.228 (0.070) | 0.169 (0.012) | 0.505 (0.030) |
| | Teacher | 0.678 (0.017) | 2.176 (0.084) | 0.158 (0.008) | 0.475 (0.021) |

clinical population.

Figure 8 further depicts transplant center–level death and graft failure rates during the COVID+KAS250 and post-pandemic periods. Each circle represents a transplant center, with both size and color intensity proportional to the observed event rate. Regional contrasts are evident: higher mortality clusters appear in the Midwest and South during the pandemic period, with noticeable improvements after 2023, whereas graft failure rates remain comparatively stable across regions. These spatial patterns, combined with the temporal trends, highlight pronounced non-stationarity in both baseline hazards and center effects, revealing the importance of adaptive modeling frameworks that can transfer, update, and recalibrate knowledge across evolving clinical regimes.

### E.3.2. TEACHER MODEL HETEROGENEITY IN ARCHITECTURES AND COVARIATES

To evaluate robustness to heterogeneous teachers, we varied the teacher model by architecture and covariate quality. Architectures included a Transformer and a linear discrete model (to mimic the coefficient-based registry models); teacher covariate quality was assessed at three levels: (a) Good: all covariates; (b) Fair: center indicators excluded; (c) Poor: only variables used in the SRTR risk-adjusted model.

Panels (a) and (b) evaluate robustness under teacher heterogeneity. Even when the teacher was weakened by covariate omission or architectural mismatch, DiSKD maintained improvements in discrimination and deviance. While Panel (a) uses a high-capacity Transformer, Panel (b) considers a simple linear hazard model—an important scenario in practice, as many clinical registries (including SRTR) publicly release only coefficient-based survival models with limited covariates. Despite the mismatch between a linear teacher and a deep student network, DiSKD continued to improve over internal-only training and over the teacher itself, indicating that prediction-level distillation can extract useful guidance without inheriting the teacher's modeling constraints.

### E.3.3. MODEL INTERPRETATION VIA VARIABLE IMPORTANCE

Beyond predictive accuracy, post-transplant risk models are often used to identify which variables most drive risk. We therefore assess the stability of SHAP-based feature-importance rankings. Panel (a) in Figure 10 reports the Jaccard overlap of the top-20 SHAP features across random seeds; DiSKD shows higher overlap for both death and graft failure, indicating more reproducible interpretations under data perturbations.

Panels (b) and (c) in Figure 10 list the top 20 predictors of one-year death and graft failure by ranking the SHAP values. Age, Kidney Donor Risk Index, and dialysis duration ranked highest, matching known prognostic factors. Other features, such as donor comorbidities and ischemia time, also aligned with clinical mechanisms. Overall, DiSKD produced variable importance patterns that were more consistent and clinically coherent than internal-only or external-only models.

*(a)* Death rate (per 100 person–months) of transplant centers in the COVID+KAS250 period.
*(b)* Death rate (per 100 person–months) of transplant centers in the post-pandemic period.

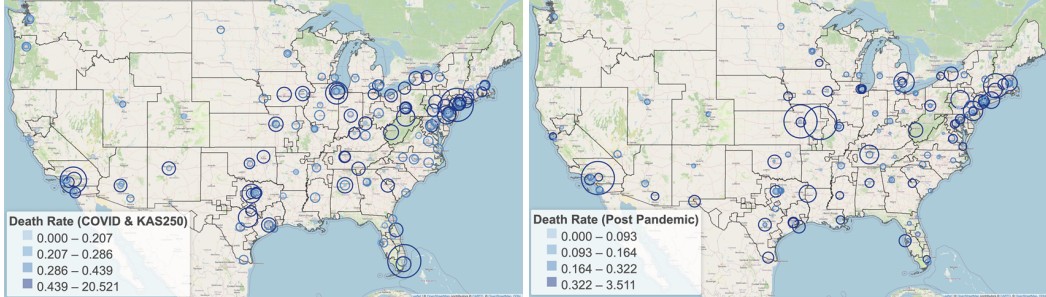

*(c)* Graft failure rate (per 100 person–months) of transplant centers in the COVID+KAS250 period.
*(d)* Graft failure rate (per 100 person–months) of transplant centers in the post-pandemic period.

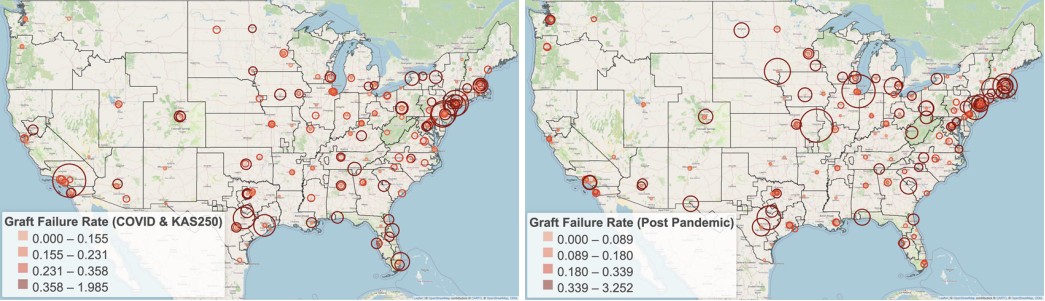

*Figure 8.* Temporal and center-level patterns in post-transplant outcomes. (a–b) Transplant center–specific death rates (per 100 person–months) during the COVID+KAS250 and post-pandemic periods, respectively. (c–d) Corresponding center-level graft failure rates for the same two periods. Each circle represents a transplant center, where both circle size and color intensity are proportional to the center's event rate. Together, the panels reveal temporal shifts and regional heterogeneity in mortality and graft failure, highlighting structural changes in transplant outcomes across centers and motivating adaptive modeling strategies.

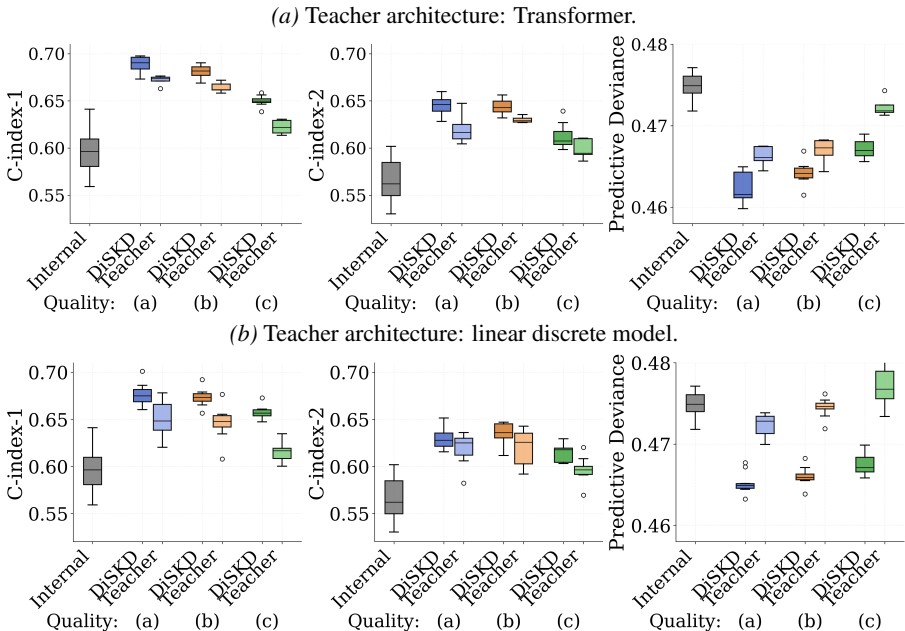

*Figure 9.* Predictive performance on the post-COVID internal cohort under different teacher modeling scenarios. All models are discrete-time competing risks models for death (risk 1) and graft failure (risk 2). Panels (a) and (b) evaluate the robustness of DiSKD with respect to teacher model architecture and covariate quality. In (a), the teacher model is a Transformer; in (b), a linear discrete model. Each architecture is trained on the COVID cohort with three covariate sets: (a) all covariates, (b) excluding transplant center indicators, and (c) using only variables selected by the SRTR risk-adjusted model. Boxplots summarize $C^{td}$ for both risks and predictive deviance, aggregated over 10 repetitions of 5-fold cross-validation. Results show that DiSKD remains robust even when teachers are weak or trained with incomplete covariates.

*(a)* Stability of variable importance across data splits (Jaccard index of top 20 SHAP features)

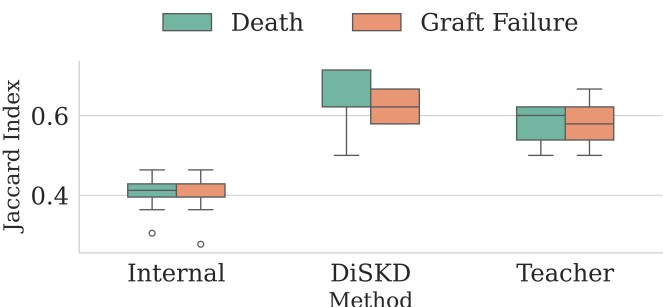

*(b)* Top 20 SHAP-ranked features for predicting one-year CIF of death (Risk 1) using DiSKD.

*(c)* Top 20 SHAP-ranked features for predicting one-year CIF of graft failure (Risk 2) using DiSKD.

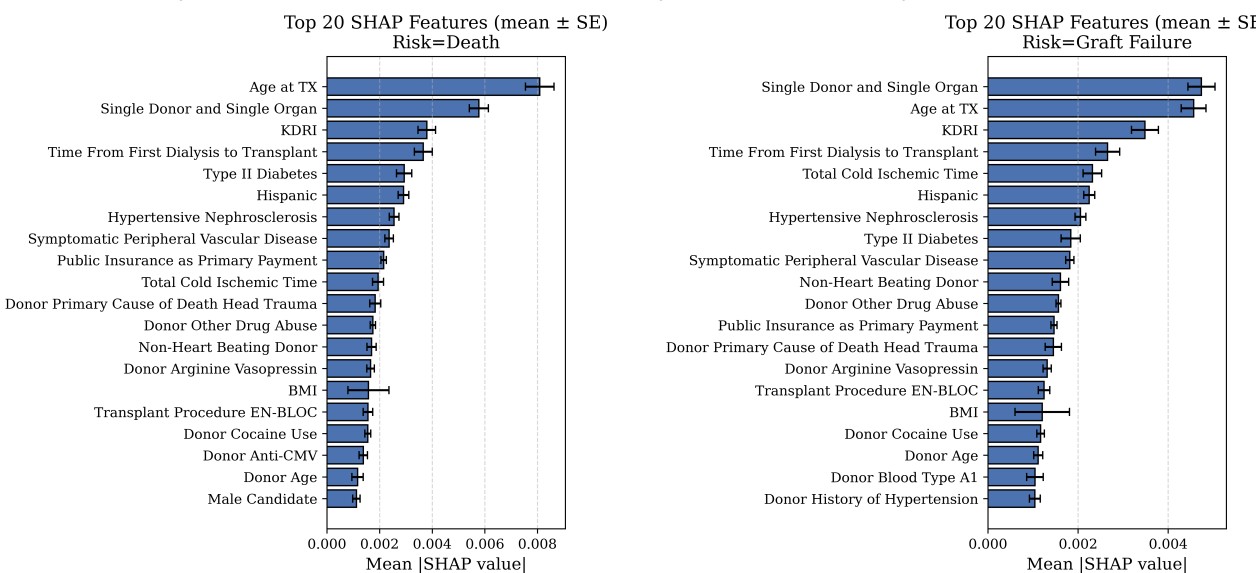

*Figure 10.* Interpretability analysis of DiSKD using SHAP values. Panel (a) shows the pairwise Jaccard index of the top 20 SHAP-ranked features across all $\binom{5}{2} = 10$ combinations of five random data splits, quantifying the consistency of variable selection for each method and risk. Panels (b) and (c) display the top 20 SHAP-ranked features for predicting the one-year CIF (i.e., at the final time point) of death and graft failure, respectively, as identified by the DiSKD model. Rankings are based on mean absolute SHAP values averaged across five random data splits, with error bars representing the standard error.

