# OpenReview forum: "Discrete Survival Knowledge Distillation for Competing Risks Analysis"
_ICML.cc/2026/Conference — ICML 2026 regular_

### Official Review · Reviewer_L6dX · 2026-03-12

**Soundness:** 3
**Presentation:** 4
**Significance:** 3
**Originality:** 3
**Overall Recommendation:** 4
**Confidence:** 4

**Summary:**

This paper proposes a knowledge distillation framework for discrete-time competing risks analysis. A larger teacher model is first trained to make predictions of the hazards and a smaller student model is then trained to match the predictive distribution produced by the teacher model. This framework is applied to improve the accuracy of risk prediction in competing risks analysis, which is more complex than standard survival analysis.

**Compliance With Llm Reviewing Policy:**

Affirmed.

**Final Justification:**

After reading through all the reviewer's comments and the authors’ responses, I decided to keep my original score and recommend a weak accept.

**Key Questions For Authors:**

See weakness.

**Limitations:**

yes

**Strengths And Weaknesses:**

Strengths:

1. This paper creatively introduces a knowledge distillation framework from machine learning into survival analysis. The framework is flexible and has the potential to be applied in many real-world scenarios.
2. The authors conducted extensive and sufficient experiments to demonstrate the effectiveness of the proposed method.
3. The paper is well written and both the methodology and experimental results are clearly presented.

Weaknesses:

1. The authors pointed a privacy-preserving advantage of the proposed framework. However, it would be helpful if they could further elaborate on how individual data privacy is protected in their implementation. In particular, if both the teacher and student models require access to the same input data when generating aligned predictions during training, it appears that the individual-level data must still be available to both models. Additional discussion on the practical privacy advantages of the method would strengthen the paper.

---

> ### Author Rebuttal · Authors · 2026-03-31
>
> We thank the reviewer for this helpful comment. We agree that the privacy-related wording in the current draft should be made more precise.
>
> The reviewer is correct that DiSKD does not by itself provide a formal individual-level privacy guarantee in the sense of differential privacy, secure multiparty computation, or cryptographic protection. It also does not eliminate the need for individual-level covariates to be available wherever teacher inference is carried out. In particular, if teacher predictions are generated on target samples during training, then the covariates required by the teacher must still be available at prediction time.
>
> The practical advantage we intended to emphasize is narrower and concerns the information that must be exchanged across institutions. DiSKD enables transfer through teacher predictions or a frozen teacher model, without requiring the teacher’s original development cohort, external outcomes, or retraining data to be shared with the target site. This is the setting emphasized in the paper, where individual-level historical data may be inaccessible and only summary information from published models or teacher predictions can be exchanged.
>
> A second and closely related advantage is that, in this weak-access setting, the external artifact is often limited and not under the analyst’s control. In practice, one may only have access to an external model defined on a different covariate set, with a different model class, or at a different outcome granularity—for example, an overall-mortality model rather than a competing-risks model, or a classical low-dimensional survival model rather than a deep neural network. If raw external individual-level data were accessible, these mismatches could often be revisited through harmonization or re-fitting. But when only summary-level external information is available, the analyst typically has no control over the outcome definition or model form used externally. DiSKD is designed for precisely this regime: it allows partial covariate overlap, accommodates diverse teacher families, and explicitly supports transfer under outcome-definition heterogeneity.
>
> More specifically, the paper develops transfer from overall-event teachers to competing-risks students, from single-risk teachers to single-risk students, and from binary-horizon teachers to competing-risks students. The empirical study also evaluates robustness to teacher misspecification and architecture mismatch, including linear teachers and teachers trained with incomplete covariates. Thus, an important practical benefit of the framework is not only reduced raw-data sharing, but also the ability to use externally available summary-level teacher information even when the external model is imperfectly aligned with the student’s target task.
>
> We will revise the paper accordingly. In particular, we will avoid presenting DiSKD as a formal privacy mechanism and instead describe it as a practical prediction-sharing transfer framework under data-sharing constraints. We will also add that an important advantage of the framework is its robustness when the externally available artifact is limited to heterogeneous summary-level teacher information, rather than harmonized external individual-level data.

---

> > ### Author Rebuttal · Reviewer_L6dX · 2026-04-02
> >
> > The authors addressed my concerns, I kept my original score.

---

> > > ### Author Response · Authors · 2026-04-06
> > >
> > > We thank the reviewer for the careful evaluation of our rebuttal and for the updated assessment.

---

### Official Review · Reviewer_6aBA · 2026-03-13

**Soundness:** 3
**Presentation:** 4
**Significance:** 4
**Originality:** 4
**Overall Recommendation:** 4
**Confidence:** 4

**Summary:**

This paper considers how to do distillation for a discrete competing risks model. They propose a temperature adjusted distillation model called DiSKD, and they show how DiSKD can be fit with a neural network by identifying a KL divergence based loss function (measuring the difference between the student and teacher models). The paper also proves that DiSKD can be modified for the following cases: outcome mismatch between the teacher and student and single target risk instead of multiple competing risks. The authors evaluate their model on synthetic data, various real survival datasets, and conduct a case study on real kidney transplant data. For the case study, they use a teacher model trained on a large historical dataset covering a period of time with two major transplant affecting disruptions: the COVID-19 pandemic and the KAS250 kidney allocation policy. They train DiSKD to match both the teacher model and a small dataset of recent patient data. They show that DiSKD performs well under outcome mismatch, teacher misspecification, covariate shift, and in both competing risks and single risk estimation settings.

**Compliance With Llm Reviewing Policy:**

Affirmed.

**Final Justification:**

This paper presents a robust model, includes extensive experimentation, and addresses real world challenges. The authors have addressed all of my concerns in their rebuttal and followup, and I accordingly increased my overall score to a 4!

**Key Questions For Authors:**

1. How does DiSKD perform with multiple teacher models?
2. How does DiSKD perform with Cox teacher models or discrete-time survival teacher models?
3. How does DiSKD perform when distilling a teacher model that predicts binary outcomes for a particular event to a student model that learns competing risks?
4. Do the authors consider any real data settings where outcome mismatch occurs? How does DiSKD perform in these settings?
5. In figure 5, how much worse is the student model to a rich model trained on donor information, recipient information, and transplant center covariates?

**Limitations:**

yes

**Strengths And Weaknesses:**

**Strengths:**

**S1: Robust model:** The authors design DiSKD to accommodate many real world complexities of medical data. For example, DiSKD accommodates diverse teachers, partial overlap in the covariate sets between teacher and student models, outcome mismatch, and data privacy constraints of the teacher model.

**S2: Extensive experimentation:** The authors run three types of experiments: (i) on simulated data, (ii) on real benchmark survival datasets, and (iii) on a real kidney transplant dataset. This is quite an extensive set of experiments!

**S3: Addressing real world challenges:** In section 4.6, the authors consider the real world challenge of distilling a rich teacher model into a simpler deployable student model. In particular, they address the real clinical constraint that organ allocation decisions can only use donor information, while the teacher model uses donor information, recipient information, and transplant center covariates.

**Weaknesses:**

**W1: Related works:** The authors should add a related works section to their draft.

**W2: External baselines:** How does DiSKD compare against external baseline models (e.g. how does it compare to a linear distillation model)?

**W3: Multiple teacher models:** DiSKD can accommodate multiple teacher models and adaptively weight information across these multiple teachers. However, no experiments are run with multiple teacher models.

**W4: Different types of teacher models:** DiSKD can accommodate diverse teacher models, including Cox models, discrete-time survival models, and deep learning models. However, experiments are only run with deep learning models.

**W5: Distilling a binary outcome model to competing risks:** The authors show in section 3.3 that DiSKD can distill a teacher model that predicts binary outcomes for a particular event to a student model that learns competing risks. However no experiments are run under this setting.

**W6: Outcome mismatch:** DiSKD’s performance under outcome mismatch is only evaluated on synthetic data. Did the authors consider any real data setting where outcome mismatch occurs?

**W7: Donor-quality prediction:** In figure 5, how much worse is the student model to a rich model trained on donor information, recipient information, and transplant center covariates? Providing this "ceiling" will give perspective on how well the student model performs.

**Minor Comments:**

- How does temperature affect the results on the real data?
- Do the authors have an explanation for why the gains in figure 5 are larger for the model with only eight features?

---

> ### Author Rebuttal · Authors · 2026-03-31
>
> We thank the reviewer for the careful reading and for highlighting several places where the empirically validated scope and the broader extensibility claims should be separated more clearly. In response, we add targeted supplementary evidence for the requested settings while keeping the paper's central empirical claim unchanged.
>
>
> **Outcome mismatch, simpler baselines, and non-deep teachers.** To address the reviewer's questions on real-data outcome mismatch, we add a compact supplementary **Table 2** (https://anonymous.4open.science/r/supp2760/Table%202.md) that consolidates these real-data slices. Specifically, on the post-COVID cohort we compare competing-risks, overall-event, and binary event-$k$ teachers distilled into a competing-risks student, and we also include a linear discrete-time competing-risks student trained with the same DiSKD objective. The pattern is consistent: all teacher granularities improve over the internal-only competing-risks student on real data; overall-event teachers are close to competing-risks teachers; binary teachers are weaker but still beneficial; and the linear student also improves substantially, showing that the gain is not tied to a high-capacity nonlinear student.
>
>
> To further separate the transfer mechanism from teacher architecture, we add a supplementary teacher-class comparison in **Table 3** (https://anonymous.4open.science/r/supp2760/Table%203.md) using both a Transformer teacher and a linear discrete-time teacher. Some of this evidence is already present in the current manuscript but dispersed across the main text and appendix; the new table is intended to make these real-data comparisons easier to inspect. The same improvement pattern holds in both cases, which directly supports robustness beyond deep teachers. We do not want to overclaim here: the evidence validates non-deep discrete-time teachers, whereas Cox-teacher support in the present submission should be understood as methodological compatibility after mapping predictions to the student's discrete hazard grid, rather than as a comprehensively benchmarked empirical claim. We will revise the wording accordingly.
>
>
> **Multiple teachers and temperature.** We agree that the multiple-teacher statement should be illustrated rather than left implicit. We therefore add a controlled two-teacher simulation with one stronger and one weaker teacher in **Table 5** (https://anonymous.4open.science/r/supp2760/Table%205.md). The learned weights behave sensibly: the stronger teacher receives the larger weight, and joint distillation stays much closer to the stronger-teacher solution while slightly improving over using that teacher alone.
>
> We also add a compact real-data temperature sensitivity analysis in **Table 6** (https://anonymous.4open.science/r/supp2760/Table%206.md). The pattern is stable and interpretable: mild softening performs best overall, while larger temperatures gradually weaken discrimination and predictive deviance, but DiSKD remains clearly better than internal-only training across the tested range.
>
>
>
> **Donor-only ceiling and Figure 5.** We agree that Figure~5 should include the richer donor+recipient+center teacher used for compression. We now report this explicit ceiling alongside the donor-only internal and donor-only DiSKD students in **Table 4** (https://anonymous.4open.science/r/supp2760/Table%204.md). Concretely, the richer teacher used for compression attains $0.6889$ C-index for death, $0.6451$ C-index for graft failure, and predictive deviance $0.4624$. This makes clear that DiSKD substantially improves the donor-only student while remaining below the richer teacher, as expected. The larger gain in the eight-feature KDRI-compatible setting is also natural: with only a small donor-only interface, the internal baseline is more information-constrained and therefore has more to gain from transfer from the richer teacher. Importantly, these donor variables are clinically grounded and available at the time of allocation, so this is a realistic deployment restriction rather than an arbitrary weakening of the student.
>
>
> **Related work and claim boundary.** We also agree that the paper should include a dedicated related-work section. In revision, we will position DiSKD more explicitly relative to transfer learning for censored outcomes, distillation methods developed for uncensored or fully specified likelihood settings, and survival-model integration approaches that do not directly address competing-risks outcome mismatch. More broadly, we will revise the manuscript so that the main claims track the empirically validated scope more tightly, while broader flexibility points are stated more carefully. We appreciate the reviewer's suggestions and believe these revisions will make the paper clearer, better balanced, and more convincing.

---

> > ### Author Rebuttal · Reviewer_6aBA · 2026-04-01
> >
> > I thank the authors for responding to most of my questions! I am accordingly increasing my overall score to 4 and my presentation score to 4. I have a follow up question for W2. I apologize if my original question was unclear. I was not asking about the performance of a linear student or linear teacher model. Rather I am curious how DiSKD compares to a transfer learning baseline.

---

> > > ### Author Response · Authors · 2026-04-06
> > >
> > > We thank the reviewer for the clarification, and we apologize for misreading W2 in our earlier response. We agree that the relevant comparison here is not a linear-teacher or linear-student ablation, but classical transfer-learning / external-model updating baselines.
> > >
> > > To address this, we added two additional baselines in the same post-COVID SRTR protocol as Figure 3(c). The first is a linear DiscreteKL baseline, i.e., a linear discrete-time transfer model. The second is a coefficient-regularization baseline, where we fit an external linear competing-risks model on the COVID+KAS250 cohort and then estimate the post-COVID target model with a penalty that shrinks the target coefficients toward the external estimates.
> > >
> > > In the updated comparison, the linear DiscreteKL baseline achieves 0.6498 (0.0182) for death C-index, 0.6196 (0.0147) for graft-failure C-index, and 0.4724 (0.0013) for predictive deviance. The coefficient-regularization baseline achieves 0.6452 (0.0144), 0.6090 (0.0090), and 0.4780 (0.0037), respectively. DiSKD remains best across all three metrics, with 0.6889 (0.0078), 0.6451 (0.0094), and 0.4624 (0.0018), respectively.
> > >
> > > We hope this better addresses the empirical intent of W2. If the reviewer had another specific transfer or updating baseline in mind, we would be glad to clarify and, if feasible within the rebuttal period, include that comparison as well.

---

### Official Review · Reviewer_VHki · 2026-03-14

**Soundness:** 2
**Presentation:** 2
**Significance:** 1
**Originality:** 2
**Overall Recommendation:** 2
**Confidence:** 3

**Summary:**

This paper introduces DiSKD, a deep learning framework designed to improve discrete-time competing risks prediction by transferring knowledge from teacher models to student models , addressing challenges such as rare events, limited sample sizes, and censored data.

**Compliance With Llm Reviewing Policy:**

Affirmed.

**Key Questions For Authors:**

Please refer to weakness

**Limitations:**

In section A.6. Theoretical Foundations, the author analysed  a low-dimensional discrete-time competing risks model. But deep learning itself is more powerful in high dimensional settings.

**Strengths And Weaknesses:**

Strength:

1. The paper is well written and easy to understand
2. The problem is interesting. Apply deep learning in survival analysis is worth researching. Computer scientists are needed for traditional statistical problems.

Weakness:

1. Regarding substantial temporal and geographic heterogeneity in kidney transplant data, the author may consider methods in phylogeography, for example, Brownian Motion Phylogeography. Why is disillation better than Brownian Motion Phylogeography? As the paper deos not have a related work section, i suggest the suthor consider add methods like these for comparison.
2. As data is lacked, and you have a teacher model, why do you want to distill teacher into a student model? Usually people want to do this because they want to reduce the inference time or they do not have enough compute for large models. Here I did not find the author is trying to sovle these problems. I suggest the author make it clear why knowledge distillation is needed.
3. Knowledge distillation without temperature can be regarded as temperature equals 1, therefore, I highly suggest authors remove the Temperature–Adjusted KL for Competing Risks section and define temperature in section Distilled Competing Risks Model. Temperature is naively assumed in knowldge distillation.
4. Beyonds forward kl used in traditional knowledge distillation, people have also found reverse kl useful in post training, for example, MiniLLM. I suggest the author to consider different divergence.

---

> ### Author Rebuttal · Authors · 2026-03-31
>
> We thank the reviewer for the thoughtful comments. We believe the main points to clarify are: (i) why knowledge distillation is needed in this setting beyond classical model compression, (ii) how temperature scaling should be positioned, (iii) whether alternative divergences are essential to the present contribution, (iv) how our problem differs from explicitly structured spatial or phylogeographic modeling, and (v) how the theory should be strengthened.
>
> First, distillation serves two primary roles in this paper: (1) Transfer and model updating: This is crucial when the target cohort is small, event-sparse, censored, and potentially shifted relative to a larger historical source cohort. In our setting, directly deploying the teacher is often suboptimal because it cannot adapt to target-cohort supervision and may suffer from population shift or mismatch in covariate availability and outcome definition. Instead, the student combines teacher guidance with the observed target data. (2) Model compression: This involves distilling a richer teacher model into a simpler, donor-only student for practical deployment. We will revise the paper to make these dual motivations more explicit.
>
> Second, we appreciate the reviewer’s point about temporal and geographic heterogeneity. However, the heterogeneity considered here is not a latent diffusion or phylogeographic problem. Rather, it enters as source--target shift in a censored competing-risks prediction task, together with possible mismatch in model form and outcome definition. Our goal is therefore not to reconstruct a spatial or evolutionary process, but to transfer predictive information from an external teacher to a local student under censoring, competing risks, and limited target-cohort data. For this reason, methods such as Brownian-motion phylogeography are not directly comparable to the prediction-transfer problem studied here. In revision, we will add a short related-work discussion clarifying this distinction.
>
> Third, our reason for including the temperature scaling is specific to the competing-risks setting: although only one event type is realized, the teacher’s full interval-outcome distribution can still encode useful cross-cause structure, and temperature provides a simple way to soften that distribution so the student is not driven only by the dominant cause. We agree, however, that this is best treated as an optional practical extension rather than a separate emphasized contribution. In revision, we will define temperature directly within the main distillation section, reduce the surrounding emphasis, and move any routine details to the appendix.
>
> Fourth, we agree forward KL is not uniquely optimal. Notably, our proposed discrete-hazard-based formulation is also a key ingredient for defining a reverse-KL objective under censoring. In revision, we will add a brief discussion of reverse-KL-type alternatives and include a supplementary comparison among KL-based distillation from teacher predictions and alternative divergence such as the Mahalanobis-type alignment objective using external summary information and a reverse-KL-type alternative. KL-based distillation operates in the weakest-access setting, requiring only teacher predictions, whereas the other two methods rely on richer external information. As shown in **Table 1** (https://anonymous.4open.science/r/supp2760/Table%201.md), all three substantially outperform internal-only training (C-index: $0.6905$, deviance: $10.0655$). Forward KL ($0.7310$ / $8.2029$) performs comparably to Mahalanobis ($0.7341$ / $8.1646$) and reverse-KL ($0.7335$ / $8.1867$). For reference, the teacher achieves $0.7281$ in C-index and $8.1186$ in predictive deviance.
>
>
> Finally, we agree with the reviewer’s limitation regarding theory of a low-dimensional aMSE analysis. In revision, we will substantially strengthen the theory by replacing the current appendix-only result with a new theorem and detailed proof for the deep discrete-time estimator itself. Specifically, we will show that the KL-integrated estimator admits a local linearization and an excess-risk expansion in which the local effect of distillation is governed by a first-order variance-reduction term and a second-order shrinkage-bias term. Under an additional remainder-stability condition used only for direct comparison on the $1/n$-shrinkage scale, we further prove a sufficient condition under which the KL-integrated estimator has strictly smaller excess prediction risk than the internal-only estimator for all sufficiently large $n$. We will move a concise statement of this theorem into the main text and keep the detailed proof in the appendix.

---

### Official Review · Reviewer_W2gw · 2026-03-16

**Soundness:** 3
**Presentation:** 2
**Significance:** 3
**Originality:** 3
**Overall Recommendation:** 4
**Confidence:** 3

**Summary:**

This paper introduces DiSKD (Discrete Survival Knowledge Distillation), a framework for transferring knowledge from a large "teacher" model to a "student" model trained on a smaller, event-sparse dataset. The work is primarily motivated by clinical settings where recent, local cohorts (e.g., post-policy change) are too small to train accurate survival models, but can benefit from the patterns learned by established models on larger, historical datasets.

Central Contributions:
* Survival-Specific KD: the method is adapted to handle typical challenges in survival analysis, specifically right-censoring and competing risks, which are not addressed by standard KD methods.
* Flexible distillation granularities: the method allows for knowledge transfer even when the teacher and student have different outcome granularities (e.g., a teacher predicting overall failure and a student predicting cause-specific mortality, or teacher predicting failure up to a certain time point).
* Empirical results: the method is demonstrated on a kidney transplant problem, where the student model achieves better calibration and discrimination on small cohorts by leveraging the teacher’s broader signal.

**Compliance With Llm Reviewing Policy:**

Affirmed.

**Final Justification:**

The rebuttal did not change my final recommendation, I think the paper has merits even though there are some caveats in writing and in making the paper widely accessible.
My review does not take into account the related concurrent submissions made by the authors, which I believe should also play a significant role in the final decision.

**Key Questions For Authors:**

Can the authors clarify the ambiguities in terms that are mentioned under weaknesses above in the "smaller comments" item?
Will the authors consider rearranging the paper to remove some of the straightforward derivations and focus on the main idea of the paper?

**Limitations:**

Yes

**Strengths And Weaknesses:**

Strengths:
* The problem is relevant and well-motivated. How to improve time-to-event model performance on small, event-sparse cohorts (e.g., recent post-policy datasets) by leveraging large-scale external "teacher" models is an important challenge in machine learning for healthcare.
* Clear Presentation: The DiSKD framework is presented in a clean and logical manner, making the adaptation of knowledge distillation to survival analysis accessible.
* Thorough Empirical Evaluation: The paper simulates real-world complexities, such as covariate shift and mismatches in outcome granularity between the teacher and student. These experiments demonstrate the practical utility of the method in heterogeneous clinical environments.

Weaknesses:
* Organizational balance and formalism: The paper spends a disproportionate amount of space detailing various adjustments to the KL divergence loss to handle different teacher-student mismatches. Given that these derivations are relatively straightforward applications of the KL definition, they could be summarized more compactly (e.g., defining the general form once and moving specific variations to the appendix). Conversely, the "Theoretical Foundations" section, which sounds like it should be an important part of the paper, is entirely relegated to the appendix and described only informally in the main text.

* Concurrent submissions: The authors disclose that there are two concurrent submissions to this conference on the same topic. While the disclosure is aprreciated, the significant overlap in contributions raises concerns about the distinct novelty of this specific submission. This point warrants consideration regarding the overall impact of the work relative to its concurrent counterparts.

* Smaller comments: The paper sometimes relies on informal terms such as "teacher-student heterogeneity" without providing formal definitions. If this term simply refers to distribution shift between the pretraining and target populations, it should be framed as such within the context of the existing OOD/Domain Generalization literature. Formalizing these concepts would significantly improve the paper's rigor; In the right column of page 3 (lines 160-161), the authors use the term "weakly identifiable." It is unclear what is meant by "weakly" in this context. Identifiability is typically a binary property of a model's ability to be recovered from infinite data; In lines 142-143, the authors state they are defining a "cause-specific" time-dependent KL loss. However, the subsequent Equation (2) does not appear to be cause-specific, as it does not index or sum over individual competing risks.

---

> ### Author Rebuttal · Authors · 2026-03-31
>
> We thank the reviewer for the careful reading and constructive suggestions. We believe the main issues are: (i) how this submission is positioned relative to the two concurrent submissions, (ii) whether the multiple KL-based formulations represent substantive modeling contributions or only straightforward algebraic variants, and (iii) clarification of terminology, Eq.(2), and the role of theory. We address these points below.
>
> First, regarding organization, we agree that the paper should better separate its central idea from routine algebra. The main contribution is not the algebraic form of several KL losses once the transfer target has been identified. Rather, the key contribution is to identify what can be validly aligned under censoring, competing risks, and heterogeneous teacher outputs. In revision, we will reorganize Sections 2--3 around one unifying principle: the teacher prediction must first be mapped to a target distribution on the student's admissible interval-outcome space, and the KL penalty is then defined on that space. The cases of competing-risks teacher, overall-event teacher, single-risk teacher, and binary-horizon teacher will be presented as instances of this principle, with more routine derivations moved to the appendix.
>
> We also agree that the theory should play a more central role. In revision, we will substantially strengthen the theoretical section by replacing the current appendix-only low-dimensional aMSE discussion with a new theorem and detailed proof for the deep discrete-time survival setting itself. Specifically, we show that the KL-integrated estimator admits a local linearization and the excess-risk expansion so that the local effect of distillation is governed by a first-order variance-reduction term and a second-order shrinkage-bias term. Moreover, under an additional remainder-stability condition, we prove that the KL-integrated estimator has strictly smaller excess prediction risk than the internal-only estimator for all sufficiently large $n$. This theorem is stronger and more directly relevant than the current appendix-only result because it is stated for the deep discrete-time estimator itself, yields a direct risk-difference formula, and gives an explicit sufficient condition for asymptotic improvement. We will move a concise statement of this theorem into the main text and keep the detailed proof in the appendix. We will also make clear that the additional remainder-stability assumption is needed only for the strict-improvement comparison, not for the leading excess-risk expansion.
>
> Second, regarding concurrent submissions, we agree that the distinction should be made more explicit and earlier in the paper. In revision, we will add a concise comparison paragraph in the introduction or related work. This submission is the only one centered on discrete-time competing risks with explicit outcome-definition mismatch between teacher and student, including overall-event $\to$ competing-risks, single-risk $\to$ single-risk, and binary-horizon $\to$ competing-risks transfer. By contrast, one concurrent submission studies distillation for locally normalized composite-likelihood objectives on matched strata, risk sets, and sampled risk sets, while the other studies transfer for deep Cox models trained through Cox-type partial likelihood. Thus, the overlap is at the level of broad motivation rather than the concrete survival setting, transfer target, or inferential structure studied here.
>
> Third, we agree that several terms should be formalized. In revision, we will define ''teacher--student heterogeneity'' as mismatch along one or more of three axes: (i) population/covariate shift between source and target cohorts, (ii) model-form or score-scale mismatch, and (iii) outcome-definition mismatch. In this paper, the primary focus is (iii), while the simulations additionally study (i) and score transformations related to (ii).  We also agree that ''weakly identifiable'' is imprecise in this context; our intended meaning was that some cross-cause structure may be weakly informed by limited internal data, and we will revise the wording accordingly.
>
> Finally, we agree with the reviewer’s comment on Eq.(2). The phrase ''cause-specific time-dependent KL'' is not the best description of that quantity. Eq.(2) is not a separate binary KL for a single cause. Rather, it is the KL divergence between the teacher and student conditional interval-outcome distributions given that the subject is still at risk at time $\tau_k$ where the possible outcomes are event type $j=1,\dots,J$ or no event in that interval. We will revise the terminology, explicitly define the interval outcome variable, and align the surrounding discussion with this interpretation.
>
> We appreciate these suggestions and believe the corresponding revisions will make the paper clearer, sharper in scope, and more rigorous.

---

> > ### Author Rebuttal · Reviewer_W2gw · 2026-04-01
> >
> > Thank you for the response, I retain my score.
> > For the concurrent submissions, I think it is difficult to assess the overlap without reading the papers. My intuition is that a single general paper on the topic of knowledge distillation in survival problems would be more useful for the community, but will leave that for discussion with the rest of the reviewers.

---

> > > ### Author Response · Authors · 2026-04-06
> > >
> > > We thank the reviewer for the follow-up comment. We understand the intuition that all three submissions concern knowledge transfer for censored time-to-event data, and that a single general paper may seem preferable. We also agree that, without reading the concurrent papers, the distinctions may not be easy to assess. To address that directly, in revision we will add a concise comparison paragraph or table early in the introduction/related work summarizing the student objective, teacher signal, transfer object, and survival-learning regime of each submission.
> > >
> > > Our main point is that the overlap is at the level of broad motivation, not at the level of the modeling object being transferred. Our submission is centered on a discrete-time competing-risks problem. The student is trained on interval-wise outcomes over multiple competing events plus "no event", and censoring already makes standard response-based KD ill-defined because censored subjects do not have a directly observed event-time target to match. On top of that, the teacher and student may use different outcome definitions: the teacher may provide full competing-risks hazards, only an overall-event hazard, only a single-risk hazard, or only a binary horizon outcome. The main methodological issue in our paper is therefore how to perform valid transfer when teacher and student do not speak the same outcome language in a censored discrete-time competing-risks setting. This is why our Section 3 is devoted to explicit mappings across outcome granularities, rather than to a single homogeneous teacher--student pairing.
> > >
> > > The component-wise paper addresses a different breakdown of standard KD. There, the issue is not competing-risks granularity, but that the student objective itself is only locally normalized, as in matched strata, risk sets, sampled risk sets, and nested case--control designs. In such settings, after conditioning, the identifiable signal is within-set relative-risk structure, not a globally normalized per-subject event distribution. Thus, standard KD fails there because the student does not live on a single global probability space that can be imitated subject-by-subject. This is why that paper defines distillation on component-induced conditional distributions over restricted outcome spaces.
> > >
> > > The deep-Cox paper studies yet another regime. Its student is a Cox-type learner based on partial likelihood and risk-set comparisons in continuous time, rather than an interval-wise competing-risks learner. The external teacher information can also be much coarser: not discrete hazard sequences, but external risk scores or rankings. Accordingly, the transfer object there is again different: the method builds the KL-type penalty on risk-set comparison distributions induced from external scores, rather than on competing-risks outcome mappings or general component-level conditional distributions.
> > >
> > > For this reason, the common appearance of a KL penalty across the three papers does not mean they are sections of one method. The KL is written on different objects because the underlying learning problems are different. In our paper, the alignment is built on interval-based competing-risks quantities and explicit mappings across outcome definitions. In the component-wise paper, the alignment is built on restricted component-level conditional distributions because globally normalized subject-level probabilities are not identified under the student objective. In the deep-Cox paper, the alignment is built on risk-set comparison distributions induced from external scores. These are therefore not cosmetic variants of one transfer rule, but distinct formulations for different censored survival-learning regimes.
> > >
> > > The same distinction appears at the level of time representation, observable structure, and empirical question. Our paper is about discrete-time competing risks with heterogeneous endpoint definitions and updating under temporal/policy shift. The component-wise paper is about design-induced local comparison problems, where conditioning and sampling determine the valid inferential target. The deep-Cox paper is about continuous-time Cox-style risk-set learning from external scores/rankings in small-data settings. Thus, the papers do not share a single student-side learning object, a single teacher-side transferable signal, or a single proof template. For that reason, a merged manuscript would be broader in topic but not more unified methodologically; it would combine different survival-learning regimes rather than present one common transfer formulation.
> > >
> > > We therefore do not view the three submissions as fragments of one larger paper. They share a high-level motivation, but they solve different methodological problems under different survival-learning structures. In that sense, we see them as complementary rather than redundant, and we will revise the paper to make that distinction easier to assess directly from this submission itself.

---

### Decision · Program_Chairs · 2026-04-30

**Decision:**

Accept (regular)

**Comment:**

While reviewers initially raised concerns about the theoretical foundations and clarity of the distillation motivation, they generally agreed that the paper presents a well-motivated and flexible framework for discrete-time competing risks. The authors' rebuttal almost addressed these issues. Most reviewers found the rebuttal convincing and maintained their positive stance. Although one reviewer remained negative, the reviewer did not follow the guidelines to acknowledge and interact in the rebuttal phase, and also did not discuss in the reviewer-AC discussion phase. Therefore, the paper is recommended for a Weak Accept.